# Utilization of palliative care services and associated factors among cancer patients in Ethiopia: A systematic review and meta-analysis

**Sadik Abdulwehab**[iD][1]*, **Frezer Kedir**[2]

**1** School of Nursing, Wollega University, Oromia, Ethiopia, **2** School of Nursing, Jimma University, Southwest Oromia, Ethiopia

* sadikabdulwehab@gmail.com

## Abstract

### Introduction

Palliative care is vital for cancer management in low- and middle-income countries like Ethiopia, but underutilization leads to unmanaged symptoms and reduced patient quality of life, and fragmented studies hinder evidence-based planning and policy development. This systematic review and meta-analysis aim to synthesize existing literature to estimate the utilization rate of Palliative care among cancer patients in Ethiopia and to identify key influencing factors.

### Review method and data sources

This study employed a systematic review and meta-analysis design to assess Palliative care utilization and its influencing factors among cancer patients in Ethiopia, sourcing evidence from various electronic databases until April 07, and studies published between 2015 and 2024 were included. The data was extracted from June 10–20 and analyzed from June 21–30, with report generation till July 27, 2025, using R software. Meta-analysis was performed using a random-effects model, with forest plots illustrating pooled prevalence and associated factors. Heterogeneity was assessed using the I² statistic, and study quality was evaluated by using a validated tool, the Joanna Briggs Institute Critical Appraisal Checklist.

### Results

A total of nine cross-sectional studies involving 2,839 cancer patients were included. The pooled Palliative care utilization rate was 42% (95% CI: 30%–54%). Educational attainment (pooled AOR = 2.57; 95% CI: 1.42–3.75) and male gender (pooled AOR = 5.58; 95% CI: 3.01–10.33) were factors significantly associated with Palliative care utilization.

**Data availability statement:** All relevant data are within the manuscript and its Supporting information files.

**Funding:** The author(s) received no specific funding for this work.

**Competing interests:** The authors have declared that no competing interests exist.

## Conclusion

This review showed the Palliative care utilization rate was 42%. Palliative care utilization in Ethiopia remains insufficient, reflecting systemic, socioeconomic, and geographic inequities. Expanding access will require decentralization of services to reach rural communities, integration of Palliative care into primary healthcare, investment in workforce capacity, and improved patient and family awareness. Strengthening these areas is essential to ensure equitable, patient-centered, and sustainable Palliative care delivery in Ethiopia.

## PROSPERO registration number

CRD420251027739.

## Introduction

Cancer remains a significant public health challenge globally, with low- and middle-income countries (LMICs) experiencing a disproportionate burden [1–3]. LMICs often lack adequate healthcare infrastructure, leading to late-stage cancer diagnoses and limited treatment options, resulting in higher mortality rates compared to high-income countries [4].

The International Agency for Research on Cancer reported in 2020, 19.3 million new cancer cases and 10.0 million cancer-related deaths globally, 70% of these deaths occurring in low- and middle-income countries [5,6]. In Ethiopia, there were 80,034 new cancer cases and 54,698 cancer-related deaths in 2022 [7]. Currently, the World Health Organization (WHO) emphasizes the importance of comprehensive care approaches that go beyond curative treatment, incorporating Palliative care [8].

Palliative care is a patient-centered, multidisciplinary approach that aims to improve the quality of life of individuals with life-threatening illnesses through early identification and management of physical, psychosocial, and spiritual needs [9]. Despite its proven benefits, Palliative care access remains severely limited worldwide, with fewer than 15% of those in need receiving care due to service shortages, lack of trained personnel, and poor integration into health systems [10,11]. These challenges are especially severe in low- and middle-income countries, where limited awareness, inadequate training, lack of structured services, cultural beliefs, geographic barriers, and socioeconomic inequities reduce Palliative care utilization [11].

In sub-Saharan Africa, Palliative care services for cancer patients remain poorly integrated and underutilized [12]. In Ethiopia, the Ministry of Health recognizes the importance of Palliative care, but its implementation and utilization remain low. This results in unmanaged symptoms, psychological distress, and increased financial and healthcare burdens for patients and families [13–15].

There is a growing yet fragmented body of evidence on the factors influencing Palliative care utilization among cancer patients in Ethiopia, as some studies underscore the role of cultural beliefs and social stigma [16], while others highlight logistical challenges such as limited access to care and a shortage of trained personnel, reflecting

inconsistencies in the literature [15]. Moreover, there is a noticeable lack of research specifically tailored to the unique Palliative care needs of Ethiopian cancer patients, particularly in areas like pain management, psychological support, and end-of-life care [17]. Compounding these challenges is the evident gap between knowledge and practice; healthcare providers often lack adequate training in evidence-based Palliative care, especially in rural regions where services are scarce [18].

Over the past two decades, Ethiopia has made significant strides in Palliative care by collaborating with the Ministry of Health, Non-Governmental Organizations, and International partners. Efforts include integrating Palliative care into the National Cancer Control Plan, establishing dedicated units, and gradually introducing oral morphine for pain management [16]. Training programs for health professionals have also been initiated to take steps toward recognizing Palliative care as a component of the health system [16,19].

Palliative care in Ethiopia faces challenges in scalability and accessibility, particularly in rural areas, where services are concentrated in urban hospitals like Addis Ababa, despite formal advancements [16]. Challenges include a limited workforce due to insufficient pre-service and in-service training, shortages in essential medications (e.g., opioids), weak community-healthcare facility linkages, and low public and provider awareness of Palliative care principles [10,20]. These issues are compounded by reliance on external donors and a lack of home-based or community-level service models [16,21].

Ethiopia has prioritized strengthening Palliative care as an essential component of Universal Health Coverage (UHC), aligning with Sustainable Development Goal 3(SDG3) and the national health sector strategies. Although Ethiopia set a target to integrate Palliative care and pain management services into at least 50% of public health facilities by 2020, recent studies show that these services remain limited, especially outside major urban centers. This gap between policy targets and actual service delivery highlights the urgent need to decentralize Palliative care, strengthen workforce capacity through pre-service and in-service training, expand community- and home-based delivery models, and better integrate Palliative care into primary healthcare to advance national UHC and SDG commitments [19,22,23].

The research on Palliative care utilization among cancer patients in Ethiopia is limited by fragmented and sometimes contradictory findings. There is no comprehensive synthesis of these studies using meta-analytic techniques to identify the most influential factors affecting Palliative care utilization in the Ethiopian context. A systematic review and meta-analysis are needed to summarize existing knowledge, quantify utilization rates, and identify consistent predictors across various settings. This evidence is crucial for national policy development and promoting equitable access to Palliative care for all.

## Methods

### Aim of the study

This systematic review and meta-analysis aim to estimate the utilization of Palliative care services and identify associated factors among cancer patients in Ethiopia to inform healthcare planning and improve Palliative care delivery.

### Design

This review employed a systematic review and meta-analysis design guided by the Preferred Reporting Items for Systematic Reviews and Meta-Analyses (PRISMA 2020) checklist [24]. Relevant studies on Palliative care utilization in Ethiopian cancer patients were identified through comprehensive database searches. Eligible studies were screened, and data were extracted using a structured form. Methodological quality was assessed using validated tools. Meta-analysis was conducted using R software, employing a random-effects model to estimate pooled proportions for utilization rates and associated factors. Heterogeneity was assessed using the I² statistic, and publication bias was evaluated using the Galbraith plot.

### Research question

This review investigates the extent of Palliative care service utilization among cancer patients in Ethiopia and identifies the socio-demographic, clinical, and system-level factors influencing it. The research question includes: What is the pooled

prevalence of Palliative care utilization among cancer patients in Ethiopia? and What are the key factors associated with utilization?

## Inclusion and exclusion criteria

Included in the review were published and unpublished observational and interventional studies reporting on Palliative care utilization and/or its influencing factors among adult cancer patients in Ethiopia. Eligible studies were required to provide at least one of the following: (a) prevalence of Palliative care utilization, or (b) quantitative data on associated factors (e.g., sociodemographic, clinical, or system-level predictors). This approach ensured that studies could meaningfully contribute to either pooled prevalence estimates or factor analysis. Studies reporting demographic or service accessibility information without utilization outcomes were excluded because they did not address our research questions. Peer-reviewed articles, theses, dissertations, and relevant gray literature were considered without language or date restrictions. Excluded from the review were case reports, expert opinions, reviews, conference abstracts, and studies lacking data on Palliative care utilization or its determinants. Non-human studies were excluded because they do not contribute to understanding patient-level service utilization. Studies focusing primarily on non-Ethiopian patients were also excluded to preserve national representativeness and ensure that findings reflect the utilization patterns of the Ethiopian population within its healthcare system.

## Search strategy

A systematic search was conducted in electronic databases, including PubMed, Scopus, Web of Science, Google Scholar, African Journals Online (AJOL), and Ethiopian university repositories. Search terms included combinations of: "Palliative care," "Palliative care utilization," "cancer," "oncology," "Ethiopia," and "associated factors." Boolean operators and Medical Subject Headings (MeSH) terms were used where applicable (Table 1). Reference lists of selected articles were also manually screened. No restrictions on publication year and language were applied. Duplicate records were removed, and two independent reviewers screened the titles, abstracts, and full texts to ensure the inclusion of eligible studies. Discrepancies were resolved through discussion or consultation with each other. The database was searched for every article published on Palliative care utilization among cancer patients till June 10, 2025, and continued to update until we sent it for publication. The data was extracted from June 10–20 and later analyzed from June 21–30, and the

**Table 1. Search strategy and retrieval summary on palliative care utilization among cancer patents in Ethiopia, 2025.**

| Database | Search Strategy (keywords/MeSH terms) | Records Retrieved | After Duplicates Removed | Full-texts Assessed | Studies Included |
|---|---|---|---|---|---|
| PubMed | ("palliative care"[MeSH] OR "palliative care utilization" OR "end-of-life care") AND ("cancer" OR "oncology") AND "Ethiopia" | 24 | 18 | 5 | 3 |
| Scopus | (TITLE-ABS-KEY ("palliative care" OR "end-of-life care") AND TITLE-ABS-KEY ("cancer") AND TITLE-ABS-KEY ("Ethiopia")) | 14 | 10 | 4 | 2 |
| Web of Science | ("palliative care" OR "end-of-life care") AND ("cancer") AND ("Ethiopia") | 12 | 9 | 3 | 1 |
| CINAHL | (MH "Palliative Care" OR "end-of-life care") AND (MH "Cancer" OR "Oncology") AND Ethiopia | 16 | 11 | 2 | 1 |
| AJOL (African Journals Online) | "Palliative care" AND "cancer" AND "Ethiopia" | 9 | 6 | 1 | 1 |
| Google Scholar | Allintitle: "palliative care" AND "cancer" AND "Ethiopia" | 16 | 12 | 1 | 1 |
| Ethiopian University Repositories | "palliative care utilization" AND "cancer" AND "Ethiopia" | 1 | 1 | 0 | 0 |
| **Total** | 82 | 65 | 16 | 9 | |

report generation was completed by July 27, 2025 (Fig 1). We registered for this study with the CRD420251027739 registration number.

## Search outcomes

A total of 82 records were identified across databases: PubMed (24), CINAHL (09), Scopus (14), Web of Science (12), CINAHL (07), and Google Scholar (16). After removing duplicates and applying inclusion criteria, 16 full-text articles were assessed. Nine studies met all inclusion criteria and were included in the final analysis. Screening and data extraction were conducted independently by two reviewers, with discrepancies resolved through discussion (Fig 1).

## Data extraction

Based on the Joanna Briggs Institute methodology principles [25], a data extraction template was created. Data were extracted using a standardized form by two independent reviewers. Extracted information included: authors, year, study design, sample size, and region. Patient-level data included age, sex, cancer type, comorbidities, and performance status. Palliative care-related data captured types of services received (e.g., pain relief, spiritual care, psychological support), utilization frequency, patient satisfaction, and barriers to access. Treatment outcomes, mortality data, and predictors of service use were also collected. Authors were contacted for missing information when necessary (Table 2).

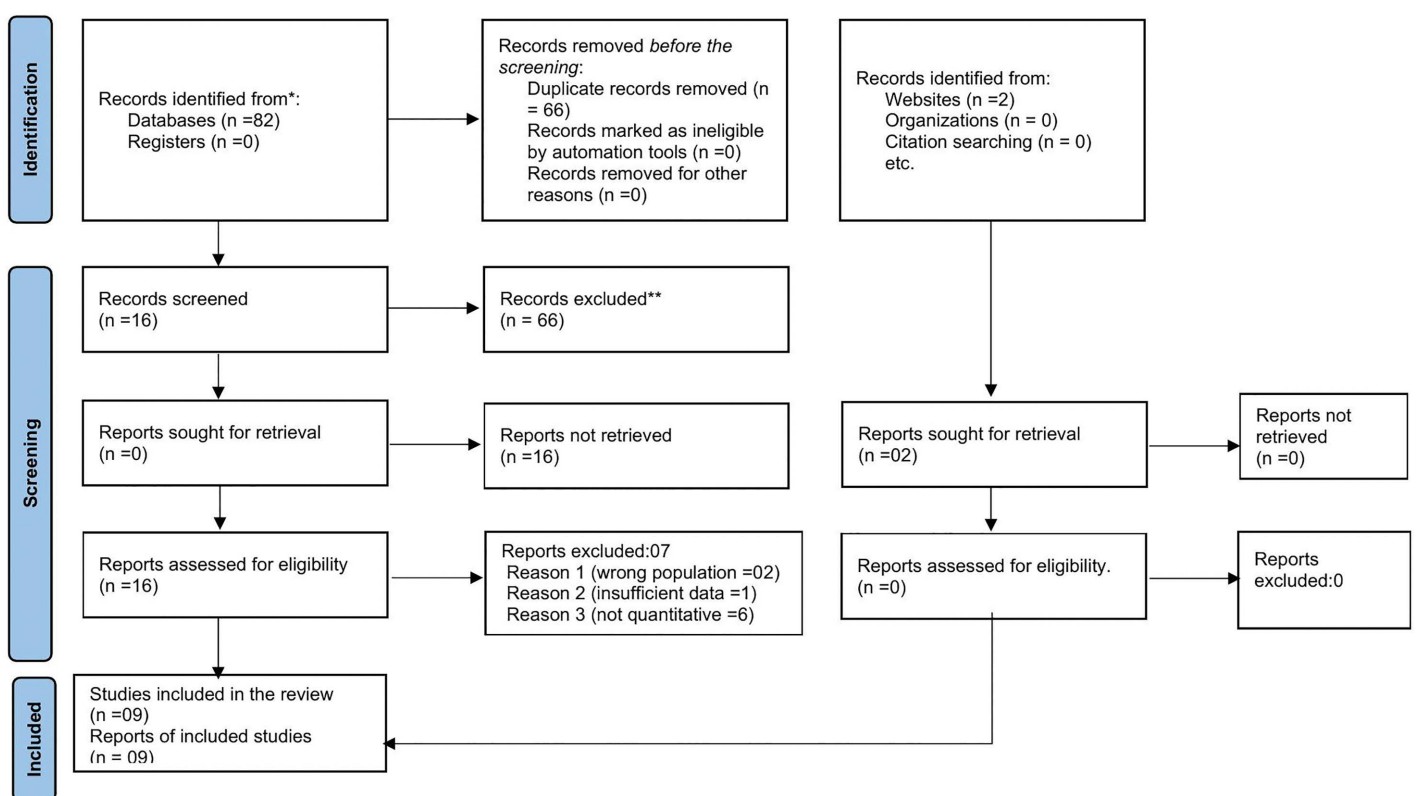

**Fig 1. Flow chart diagram and PRISMA checklist describing the selection of studies for the systematic review and meta-analysis on Palliative care among cancer patients in Ethiopia, 2025.**

**Table 2. Characteristics of included studies on Palliative care utilization and its influencing factors among cancer patients in Ethiopia, 2025.**

| Author & year | Title | Region/ Setting | Design | Sample & Population | Characteristics | Types of PC Given | Outcome Measures | PC Delivery | Utilization Rate | Associated Factors with AOR, CI, P value | Limitations | Conclu- sions | Recommen- dations |
|---|---|---|---|---|---|---|---|---|---|---|---|---|---|
| Afessa et al. - 2024 | Pallia- tive care service utilization and associ- ated factors among can- cer patients at oncology units of public hospitals in Addis Ababa, Ethiopia | Addis Ababa (Tikur Anbessa & St. Paul) | Institution- based cross- sectional | 404 adult cancer patients | 63.4% aged 18–47, 56.7% female, 63.4% urban | Symptom manage- ment, psy- chological support, end-of-life care | PC utilization (35.4%), knowledge, attitude, satisfaction | Public hospital oncology units | 35.4% [95% CI: 31.4– 40.3%] | Education (college/ university): AOR=2.3, 95% CI: 1.15–4.61, p<0.05 Living <23 km from facility: AOR=1.8, 95% CI: 1.07–3.09, p<0.05 Inability to read/ write: AOR=0.31, 95% CI: 0.14–0.67, p<0.05 Low satisfaction with healthcare: AOR=0.40, 95% CI: 0.23–0.72, p<0.05 Treatment side effects: AOR=0.41, 95% CI: 0.24–0.72, p<0.05 | Only urban settings; cross- sectional design | Low PC utilization; systemic/ personal barriers | Enhance education, accessibility, and integra- tion of PC |
| Amare et al. - 2023 | The Prevalence of Cancer Patients Requiring Palliative Care and Its Associated Factors at St. Paul Hospital | St. Paul Hospital, Addis Ababa | Cross- sectional | 301 admit- ted cancer patients | Mean age 42, 58.5% female, from various Ethiopian regions | Not directly delivered; need assessed via SPICT-LIS | 10.6% needed PC, symptoms burden, per- formance decline | Inpatient oncology ward | 10.6% (need identified) | Age>61: AOR=2.39, 95% CI: 0.34–16.55 Male gender: AOR=5.31, 95% CI: 1.68–11.79, p=0.001 Education (no formal education): Associated with higher PC need, p=0.01 Region (e.g., SNNPR 25% vs Addis Ababa 0%): p<0.0001 Marital status (mar- ried): More likely to need PC, p=0.005 | Single- center; tool-based assessment | High unmet need for PC among inpatients | Early PC screening; strengthen hospital- based services |
| Fetene et al., 2024 | Factors affecting need and utilization of palliative care services among Ethiopian women in an oncology department | Hawassa University Compre- hensive and Spe- cialized Hospital, Sidama Region | Hospital- based cross- sectional study | 121 women ≥18 years with breast cancer | Mean age 50, 58.7% rural, 50.4% primary schoo, majority stage 3 BC | Counsel- ing, psy- chother- apy, pain treatment, chemo- therapy | Utilization categorized as better or worse based on mean score | Pro- vided by physicians &nurses in inpatient &outpatient units | 59.5% worse utilization (no CI reported) | Rural residence (AOR=11.82, p<0.05) | Small sam- ple size, sin- gle setting, no control group | More than half had poor PC utilization; rural resi- dence was a major factor | Expand PC access and outreach in rural communities |

*(Continued)*

| Author & year | Title | Region/ Setting | Design | Sample & Population | Characteristics | Types of PC Given | Outcome Measures | PC Delivery | Utilization Rate | Associated Factors with AOR,CI, P value | Limitations | Conclusions | Recommendations |
|---|---|---|---|---|---|---|---|---|---|---|---|---|---|
| Kebebew et al. - 2022 | Hospital-based evaluation of palliative care among patients with advanced cervical cancer | Radiotherapy center, Specialized tertiary hospital, Ethiopia | Cross-sectional, hospital-based | 385 women with advanced cervical cancer | Mean age: 52 years; 61.8% aged ≥50; 63.1% illiterate; 84.2% with income<$50/month | Pain treatment, symptom control (bleeding/discharge), limited counselling, spiritual, economic & home-based care | Knowledge, attitude, practice of PC; pain control; access to support | Mostly hospital-based, with some home visits | ~90.3% received pain treatment, but only 56.3% had fair or complete relief | No AOR/CI/p–values reported for PC utilization or factors; analysis was descriptive | Excluded early-stage patients and those not attending the hospital; focus only on advanced cases | Most received some PC, but it was non-comprehensive; poor in education, spiritual and psychological support | Raise awareness of full PC scope; provide culturally appropriate communication on death & dying; ensure comprehensive PC at all healthcare levels |
| Lakew et al., 2015 | Assessment of knowledge, accessibility and utilization of palliative care services among adult cancer patients | Tikur Anbessa Specialized Hospital, Addis Ababa | Institution-based cross-sectional study | 384 adult cancer patients | Mean age 45.8, 62% married, 89.8% reported accessibility issues | Hospital counselling, community support, home-based care | Utilization based on basic PC service responses | Delivered through inpatient and outpatient departments | 69% utilization (no CI reported) | Knowledge (AOR=26.9, CI: 12.3–59); Little physical wellbeing (AOR=3.1, CI: 1.96–4.9); Full social wellbeing (AOR=1.7, CI: 1.01–2.8); Income $25–50 (AOR=0.25, CI: 0.09–0.7); Single marital status (AOR=55.4, CI: 1.2–2660.4) | Self-reporting, recall and social desirability bias | Knowledge and wellbeing influenced utilization; access remains limited | Introduce educational sessions, improve service awareness and support |
| Teklemariam et al., 2022 | Perception about palliative care and factors influencing the likelihood of palliative care service utilisation | Tikur Anbessa Specialized Hospital, Addis Ababa | Facility-based cross-sectional study | 304 adult cancer patients | Median age 56, 59.5% female, 56.9% rural, 48.4% no formal schooling | Chemotherapy, radiation, complaint therapy, supportive PC | Utilization determined via Likert scale adjusted score | PC services from hospital-based oncology team | 42.8% utilization (no CI reported) | Income ≥$52.35 (AOR=2.36, CI: 1.37–4.06); Family members >2 (AOR=2.28, CI: 1.02–5.13); Gov't employee (AOR=0.42, CI: 0.20–0.87); Formal schooling (AOR=0.51, CI: 0.23–0.94) | Cross-sectional design limits causality; single facility | Utilization remains low; affected by income, family support, &employment status | Design targeted outreach for low-income and less-supported patients |

(Continued)

| Author & year | Title | Region/ Setting | Design | Sample & Population | Characteristics | Types of PC Given | Outcome Measures | PC Delivery | Utilization Rate | Associated Factors with AOR,CI, P value | Limitations | Conclusions | Recommendations |
|---|---|---|---|---|---|---|---|---|---|---|---|---|---|
| Worku et al., 2017 | Rehabilitation for cancer patients at Black Lion hospital, Addis Ababa, Ethiopia: a cross-sectional study | Black Lion Hospital, Addis Ababa | Cross-sectional study | 388 adult cancer patients | Mean age 44, 68.6% female, 25% breast cancer, 20.6% colorectal | Rehabilitation, nutritional and psychosocial support | Service use recorded if received at least once | Rehabilitation services by hospital-based team | 26% utilization (no CI reported) | Not specified in detail | Convenience sampling, missing CI and detailed factor | Low rehabilitation service utilization among patients | Enhance access, train staff, and increase resource availability |
| Bunare et al., 2022 | Utilization of Rehabilitation Services and Associated Factors Among Adults With Cancer Diagnoses | Hawassa Comprehensive Specialized Hospital, Ethiopia | Institutional-based cross-sectional study | 325 adults with cancer diagnoses selected via systematic sampling | Adults with cancer; both genders; assessed on ADL, social support, satisfaction | Cancer rehabilitation (physical, cognitive, emotional, social support) | Utilization defined as attending ≥1 rehabilitation service in the last year | Hospital-based rehabilitation services | 33.2% (95% CI: 27.93–41.25) | - Male (AOR = 5.76; 95% CI: 2.60–12.75) – Urban residence (AOR = 2.56; 95% CI: 1.04–6.26) – Independent in ADLs (AOR = 2.68; 95% CI: 1.20–6.00) – Received education on rehab (AOR = 2.44; 95% CI: 1.21–4.91) – Strong social support (AOR = 2.10; 95% CI: 1.02–4.87) – Satisfaction with care (AOR = 3.21; 95% CI: 1.42–5.76) | Based on self-report and medical records; institutional scope may limit generalizability | Only one-third of patients utilized rehabilitation; several demographic and psychosocial factors influenced use | Improve awareness through patient education and strengthen supportive systems to enhance utilization |
| Aynalem et al., 2023 | Utilization of palliative care and its associated factors among adult cancer patients in Hawassa University Comprehensive Specialized Hospital oncology center, Hawassa, Ethiopia, 2021 | Hawassa University Comprehensive Specialized Hospital, Oncology Center | Institution-based cross-sectional study | 180 adult cancer patients (≥18 years), randomly selected | 66% were aged ≥50 years; income, education, and accessibility varied | Not explicitly listed (general palliative care services provided by the oncology center) | Categorized as "better" utilization based on structured questionnaire data | PC services delivered within oncology unit at the hospital | 63% utilization (CI not explicitly provided) | - Age <50 (AOR = 2.7; 95% CI: 1.13–6.63) – Education (Grade 9–12: AOR = 1.46; CI: 0.41–5.21; College+: AOR = 3.23; CI: 0.98–10.61) – Income >5,500 Birr (AOR = 2.7; CI: 0.51–5.76) – Good accessibility of PC (AOR = 2.99; CI: 1.21–3.28) | Limited sample size; cross-sectional design limits causal inference; CI not consistently precise | Two-thirds of patients had better PC utilization; older, less-educated, low-income, and rural patients had lower access | Improve PC information provision, especially for older and less-educated patients; enhance accessibility in rural & suburban areas |

## Organizing, summarizing, and reporting the results

Findings were organized into three main categories: (1) Demographic and clinical characteristics of cancer patients receiving Palliative care, (2) Utilization patterns of Palliative care services and the types provided, and (3) Predictors of Palliative care service use. Meta-analyses were conducted to pool estimates of Palliative care utilization rates and influencing factors. Results were reported in descriptive and tabular formats and illustrated using charts and figures. The PRISMA checklist guided the reporting process to ensure rigor and transparency.

## Quality appraisal

The methodological quality of the included studies was evaluated using the Joanna Briggs Institute (JBI) Critical Appraisal Checklist for Analytical Cross-Sectional Studies [25]. Each study was independently assessed by two reviewers across domains such as sample selection, outcome measurement, control of confounding variables, and statistical analysis. Studies scoring 7–8 "Yes" responses were considered low risk of bias, 4–6 as moderate risk, and 0–3 as high risk. Only low- and moderate-risk studies were included in the final synthesis (Table 3).

## Statistical analysis

Data analysis was performed using R software. Descriptive statistics, including frequencies, percentages, means, and standard deviations, were used to summarize study characteristics.

A meta-analysis was conducted using a random-effects model. A random-effects approach using Restricted Maximum Likelihood (REML) estimation was selected a priori due to anticipated clinical and methodological heterogeneity across studies. This approach estimates the mean of a distribution of true effects rather than assuming a single common effect size. Publication bias and small-study effects were explored using the Galbraith (radial) plot. Although alternative approaches, such as the Doi plot and LFK index, have been recommended for prevalence meta-analyses, a valid implementation could not be obtained using available R for the current dataset. In accordance with recent guidance, we employed REML estimation and applied Hartung-Knapp adjustments to compute more robust confidence intervals for

Table 3. JBI critical appraisal on Palliative care service utilization among cancer patients in Ethiopia, 2025.

| Study No. | Afessa et al., 2024 | Amare et al., 2023 | Kebebew et al., 2022 | Lakew et al., 2015 | Teklemariam et al., 2022 | Worku et al., 2017 | Bunare et al., 2022 | Aynalem et al., 2023 | Fetene et al., 2024 |
|---|---|---|---|---|---|---|---|---|---|
| Was there a clear statement of the aims of the research? | ✓ | ✓ | ✓ | ✓ | ✓ | ✓ | ✓ | ✓ | ✓ |
| Was the study design appropriate for the aims of the research? | ✓ | ✓ | ✓ | ✓ | ✓ | ✓ | ✓ | ✓ | ✓ |
| Was the sample representative of the population studied? | ✓ | ✗ | ✓ | ✓ | ✓ | ✗ | ✓ | ✓ | ✓ |
| Was the sample size adequate? | ✓ | ✗ | ✗ | ✓ | ✓ | ✓ | ✗ | ✗ | ✗ |
| Were the study subjects and the setting described in detail? | ✗ | ✓ | ✓ | ✓ | ✓ | ✓ | ✓ | ✓ | ✓ |
| Was the data collected in a reliable way? | ✗ | ✓ | ✗ | ✗ | ✓ | ✓ | ✓ | ✓ | ✓ |
| Were the statistical analyses used to assess the data appropriate? | ✓ | ✓ | ✓ | ✓ | ✓ | ✓ | ✓ | ✓ | ✓ |
| *Were the findings valid and applicable to the local context?* | ✓ | ✓ | ✓ | ✓ | ✓ | ✓ | ✓ | ✓ | ✗ |
| *Overall Quality* | 6 | 6 | 6 | 7 | 8 | 7 | 7 | 7 | 6 |

Key: ✓ = Yes (Criterion met), ✗ = No (Criterion not met) and ? = Unclear (Not adequately reported).

pooled estimates and heterogeneity parameters [26,27]. Hartung–Knapp adjustments were applied to produce more conservative confidence intervals, particularly important given the small number of studies and high heterogeneity.

Heterogeneity was assessed using I² statistics, with substantial heterogeneity defined as I² > 50%. We also explored random-effects models for aggregate data as a sensitivity approach [28]. Given the expected variability in study populations, settings, and measurement approaches, pooled prevalence estimates were interpreted cautiously as descriptive aggregates rather than definitive national estimates. Sensitivity analyses were performed by excluding studies at high risk of bias or with small sample sizes. A leave-one-out sensitivity analysis was performed using a random-effects model using the Tau² value, which represents the estimated between-study variance under the random-effects model, reflecting the magnitude of true heterogeneity beyond sampling error. All findings were reported with 95% confidence intervals, with statistical significance set at p < 0.05. This reporting conforms with the PRISMA 2020 guidelines, and we have also considered emerging enhancements expected in PRISMA 2025 [24,29].

### Certainty of evidence assessment

The certainty of evidence was assessed using the Core GRADE approach, which provides a streamlined evaluation of overall certainty following meta-analysis, particularly suitable for observational studies.

### Ethical consideration

This review involved a secondary analysis of existing published studies, and therefore, ethical approval was not required. Ethical principles were upheld by ensuring all included studies had appropriate ethical clearance and informed consent. Proper citation and acknowledgment of original authorship were maintained throughout to ensure transparency and respect for intellectual property.

## Results

### Characteristics of the included studies

The included studies were all cross-sectional in design and conducted within various institutional settings across Ethiopia, primarily in public hospitals and specialized oncology centers [13,14,30–36]. Sample sizes ranged from 121 to 404 adult cancer patients, with participants mostly aged between 18 and 66 years and a majority being female [14,31,35]. Most studies recruited participants from urban hospitals in Addis Ababa, such as Tikur Anbessa and St. Paul's [13,14,31,33–35], while others included patients from Hawassa and Sidama regions to reflect a more diverse geographic representation [30,32,36].

### Types of palliative care provided for cancer patients in Ethiopia

Across the reviewed studies, Palliative care services for cancer patients in Ethiopia varied in scope, delivery, and comprehensiveness. The most commonly reported components included pain and symptom management, psychological support, and end-of-life care [14,30]. In some cases, care was extended to include counseling, chemotherapy, radiation, and complaint-based supportive care [13,31]. At specialized units, such as those in Hawassa and tertiary hospitals, services incorporated psychotherapy, economic assistance, and limited spiritual support, although these were often inconsistently applied or underdeveloped [33,36]. Community and home-based care were mentioned in a few studies, but were not widespread or fully integrated [30,33]. Rehabilitation-focused studies highlighted nutritional, cognitive, emotional, and social support services, indicating efforts to expand Palliative care beyond clinical symptom control [32,34].

### Palliative care service delivery among cancer patients in Ethiopia

Palliative care delivery across the included Ethiopian studies was predominantly institutional, with services administered within hospital-based oncology units or inpatient and outpatient departments [13,14,30]. Two studies [30,33] referenced

community or home-based services. Palliative care delivery models largely lacked specialized structures or referral systems and were primarily focused on curative or hospital-based treatment settings, rather than holistic end-of-life care. In some instances, such as in Hawassa and Addis Ababa, the delivery mechanisms were limited to inpatient hospital units, restricting accessibility for rural populations and those with mobility or economic challenges [31,36].

### Palliative care service measures

The studies included in this review utilized a range of outcome measures to assess Palliative care utilization and effectiveness among cancer patients in Ethiopia. Most commonly, outcomes focused on the rate of Palliative care utilization are typically reported as a percentage of participants receiving or needing Palliative care services [14,36]. Several studies also assessed patient knowledge, attitude, and satisfaction toward Palliative care as indirect indicators of service uptake and perception [14,30]. A functional performance decline, symptom burden, and the need for Palliative care services were evaluated as Palliative care utilization [35]. Other studies categorized utilization levels based on structured questionnaire scores as better or worse [31,36]. For rehabilitation-related services, outcomes included attendance at rehabilitation sessions, level of independence in activities of daily living (ADLs), and patient satisfaction with care [32,34]. Despite this diversity, the lack of standardized and validated outcome measures across studies made cross-comparison challenging [13,14,30–36].

Palliative care utilization was measured variably across studies, including receipt of at least one Palliative care service, referral to a Palliative care unit, self-reported utilization, or classification based on structured questionnaire scores. Although this variability reflects real-world practice, it contributes to between-study heterogeneity.

### Palliative care service utilization rate among cancer patients in Ethiopia

Palliative care utilization rates among cancer patients in Ethiopia showed marked variability across the included studies, reflecting heterogeneity in service availability, health system readiness, and patient-level factors. The reported prevalence ranged from 10.6% in Amare et al. (2023 to 69.0% in Lakew et al. (2015. The random-effects meta-analysis, including nine studies, estimated a pooled prevalence of 42% (95% CI: 30%–54%). Given the substantial heterogeneity observed across studies, this pooled estimate should be interpreted as a descriptive aggregate rather than a definitive prevalence (Fig 2).

### Subgroup analysis

To investigate potential sources of heterogeneity, subgroup analysis was conducted by geographic region. In Addis Ababa, the pooled prevalence of palliative care utilization was 39.0% (95% CI: 23%–57%), based on six studies, while outside Addis Ababa, the pooled prevalence was slightly higher at 46.0% (95% CI: 28%–63%), based on three studies. Although utilization rates appeared somewhat higher outside Addis Ababa, the differences were not statistically significant, suggesting that regional location alone may not account for the observed variation. Heterogeneity remained high within both subgroups ($I^2 = 98.79\%$ for Addis Ababa; $I^2 = 94.97\%$ for outside Addis Ababa) (Fig 3). All pooled analyses demonstrated substantial heterogeneity ($I^2 > 90\%$), indicating considerable variability across studies. Consequently, pooled estimates should be interpreted as indicative rather than definitive national values.

The presence of publication bias was visually assessed using a funnel plot. The Galbraith plot did not reveal marked asymmetry or influential outliers, suggesting no strong evidence of publication bias among the included studies. However, in this meta-analysis, there is no strong evidence of publication bias noted in the data summary (Fig 4).

### Sensitivity analysis

A leave-one-out sensitivity analysis was performed using a random-effects model ($Tau^2 = 0.0067$) to evaluate the influence of individual studies on the overall pooled estimate. The analysis revealed only slight variation in the pooled effect size

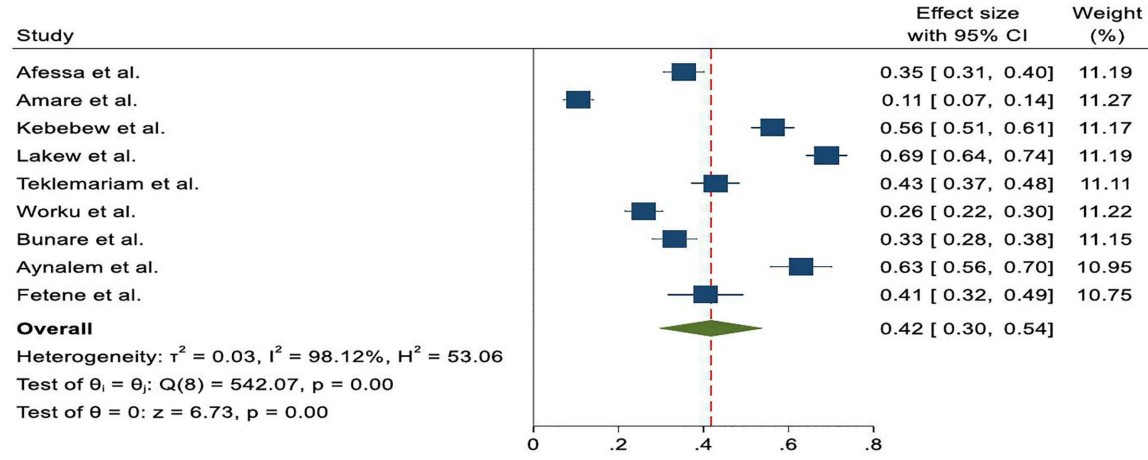

**Fig 2. Forest plot shows the pooled utilization rate of Palliative care service among cancer patients in Ethiopia, systematic review and meta-analysis, 2025.**

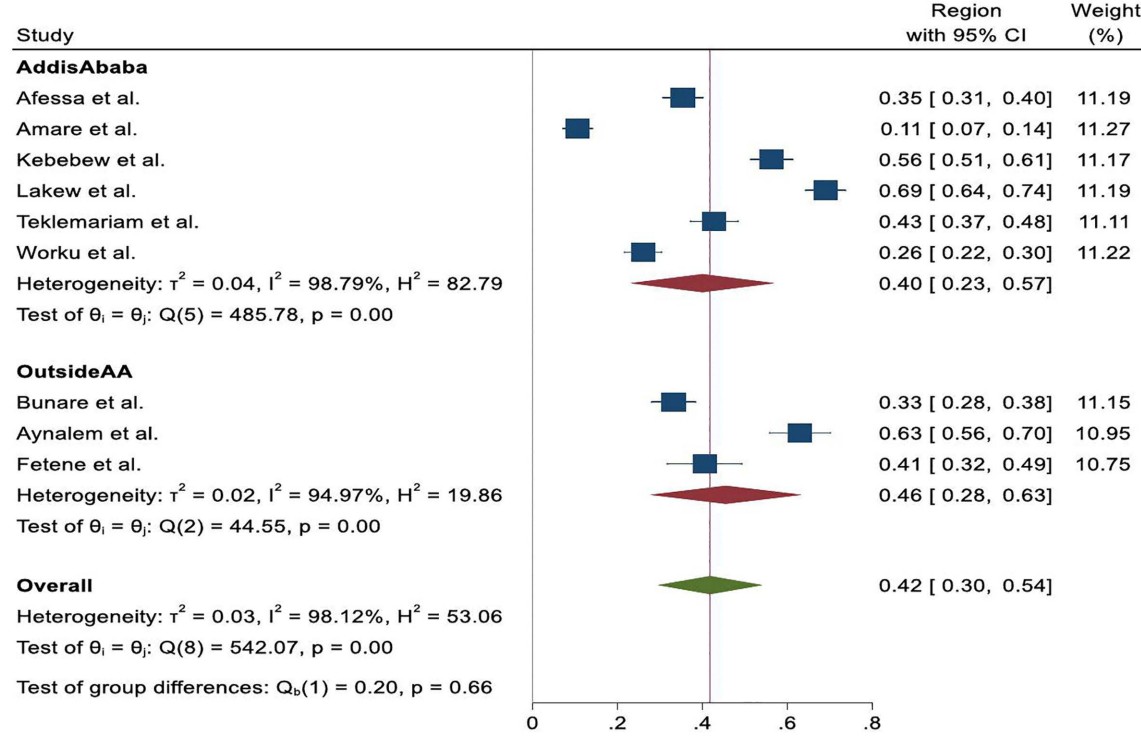

**Fig 3. Forest plot indicating sub-analysis of utilization of Palliative care based on region among cancer patients in Ethiopia, systematic review and meta-analysis, 2025.**

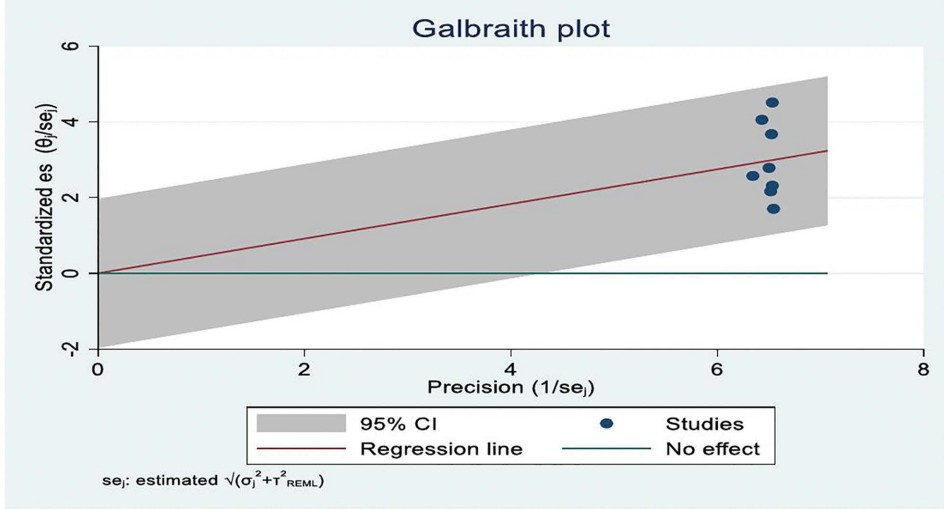

**Fig 4. Galbraith plot shows pooled utilization rate of Palliative care service among cancer patients in Ethiopia, systematic review and meta-analysis, 2025.**

when each study was excluded one at a time, with estimates ranging from 0.380 to 0.457. The confidence intervals consistently overlapped, suggesting that no single study exerted a significant influence on the overall findings (Table 4).

## Factors associated with palliative care utilization among cancer patients in Ethiopia

Included studies varied in the way they reported associations between predictors and Palliative care utilization. For studies that did not provide adjusted or crude odds ratios, raw frequency data were extracted where possible and used to compute crude odds ratios with 95% CIs. When raw data were insufficient for calculation, findings were narratively synthesized and described qualitatively. This ensured that all eligible studies contributed to the review, while avoiding potential bias from imputing unavailable estimates. Predictors reported by fewer than two studies with comparable effect measures were synthesized narratively and interpreted cautiously.

Factors associated with Palliative care utilization among cancer patients were assessed across the included articles in Ethiopia. Given the analysis of multiple factors, results should be interpreted cautiously, as the risk of type I error

**Table 4. Influence of each study on pooled effect size on palliative care service utilization among cancer patients in Ethiopia, 2025.**

| Omitted Study | Region | Pooled Effect Size | 95% CI Lower | 95% CI Upper |
|---|---|---|---|---|
| Afessa et al., 2024 | Addis Ababa | 0.423 | 0.363 | 0.483 |
| Amare et al., 2023 | Addis Ababa | 0.457 | 0.397 | 0.517 |
| Kebebew et al., 2022 | Addis Ababa | 0.396 | 0.336 | 0.456 |
| Lakew et al., 2015 | Addis Ababa | 0.380 | 0.320 | 0.440 |
| Teklemariam et al., 2022 | Addis Ababa | 0.414 | 0.354 | 0.474 |
| Worku et al., 2017 | Addis Ababa | 0.436 | 0.375 | 0.496 |
| Bunare et al., 2022 | Hawassa | 0.426 | 0.366 | 0.486 |
| Aynalem et al., 2023 | Hawassa | 0.390 | 0.331 | 0.450 |
| Fetene et al., 2024 | Hawassa | 0.416 | 0.357 | 0.476 |

increases with multiple comparisons. Age was identified in one study as influencing Palliative care utilization, with patients under 50 years more likely to use Palliative care services by Aynalem et al. [36](AOR = 2.7, 95% CI: 1.13–6.63). Similarly, experiencing treatment side effects was reported in another study to be inversely associated with Palliative care utilization (Afessa et al., 14); AOR = 0.41, 95% CI: 0.24–0.72, p < 0.05). These findings are based on individual studies and interpreted cautiously.

Educational status appears to have a mixed influence. Higher education (college or university level) was positively associated with Palliative care use in several studies [14,32,36]. Pooled effect size for the association between educational status and Palliative care utilization among cancer patients in Ethiopia was found to be 2.49(CI: 2.04–2.95) (Fig 5).

The study found that the Galbraith plot did not reveal marked asymmetry or influential outliers, suggesting no strong evidence of publication bias among the included studies (Fig 6).

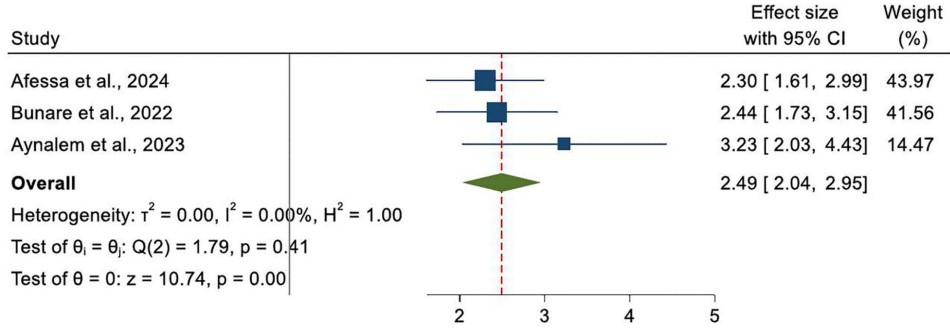

**Fig 5. Forest plot shows the association of educational status and Palliative care service among cancer patients in Ethiopia, systematic review and meta-analysis, 2025.**

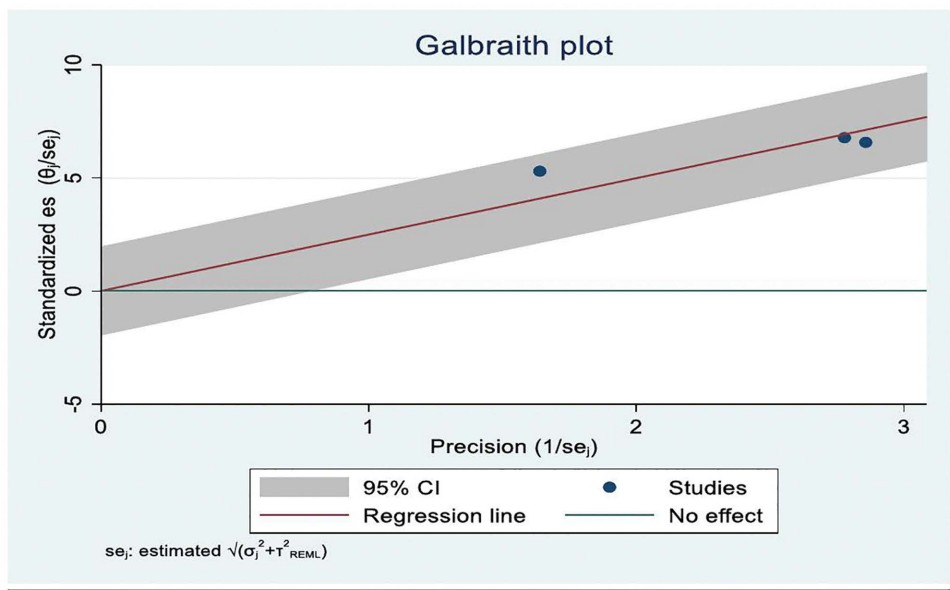

**Fig 6. Galbraith plot shows the association of educational status and Palliative care service utilization among cancer patients in Ethiopia, systematic review and meta-analysis, 2025.**

Gender was identified as a significant factor influencing Palliative care utilization among cancer patients in Ethiopia. Two studies [23,26] demonstrated a strong positive association between being male and higher service use. Bunare et al. [32] reported that male patients had markedly greater odds of utilizing Palliative care compared to females (AOR = 5.76; 95% CI: 2.60–12.75), while Amare et al. [35] found a similar association (AOR = 5.31; 95% CI: 1.68–11.79). When the findings from these two studies were pooled, the combined effect indicated that male patients were significantly more likely to access Palliative care services than their female counterparts (pooled AOR = 5.58; 95% CI: 4.97–6.20) (Fig 7).

A forest plot was generated to display the individual and pooled adjusted odds ratios for the association between sex and Palliative care utilization. Because the pooled analysis was based on only two studies, assessment of publication bias using funnel plots or statistical tests was not performed.

Access to healthcare services was identified as an important determinant of Palliative care utilization. One study reported that patients living within 23 kilometers of a healthcare facility had higher odds of using Palliative care services (Afessa et al.; AOR = 1.80) [14]. while another study found that patients with good accessibility to services were more likely to utilize Palliative care (Aynalem et al; AOR = 2.99) [36].

Satisfaction with care emerged as an important determinant of Palliative care utilization. Bunare et al. [32] reported that satisfied patients were significantly more likely to use services (AOR = 3.21; 95% CI: 1.42–5.76), while Afessa et al. [14] similarly found a positive association (AOR = 1.40; 95% CI: not reported); however, this estimate was not included in the meta-analysis because the confidence interval or standard error was not reported.

In addition, family and social support were consistently linked with higher service use. Teklemariam et al. [13] reported that patients with stronger family involvement had greater odds of utilizing services (AOR = 2.28; 95% CI: 1.02–5.13), and Bunare et al. [32] reported that patients with strong social support had greater odds of utilizing services (AOR = 2.10; 95% CI: 1.02–4.87).

Using the Core GRADE framework, the certainty of evidence for pooled Palliative care utilization and associated factors was rated as low to very low, primarily due to high heterogeneity, cross-sectional study designs, and inconsistency across studies. Bayesian meta-analysis was not performed due to the small number of studies and limited data for robust estimation. Although funnel plots and Egger's tests were performed, the small number of studies (<10 per factor) limits their reliability; thus, the absence of detected publication bias does not confirm that bias is absent.

Given the analysis of multiple factors, results should be interpreted cautiously, as the risk of type I error increases with multiple comparisons. Overall, these findings indicate that sociodemographic factors, economic capacity, healthcare accessibility, patient satisfaction, and social support play important roles in influencing Palliative care utilization among

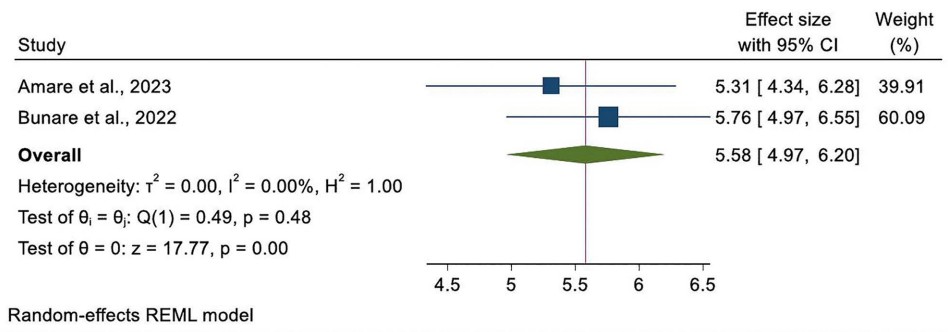

**Fig 7. Forest plot shows the association of educational status and Palliative care service utilization among cancer patients in Ethiopia, systematic review and meta-analysis, 2025.**

Ethiopian cancer patients and are summarized in the table as a summary of factors associated with Palliative care Utilization among Cancer patients in Ethiopia (Table 5).

## Certainty of evidence assessment

The certainty of evidence was assessed using the Core GRADE approach, which provides a streamlined evaluation of overall certainty following meta-analysis, particularly suitable for observational studies (Table 6).

## Discussion

This systematic review and meta-analysis synthesized evidence from nine cross-sectional studies assessing Palliative care service utilization among adult cancer patients in Ethiopia [13,14,30–36]. The overall findings demonstrate that

**Table 5. Summary of factors associated with Palliative care utilization among cancer patients in Ethiopia.**

| Factor | No. of Studies | Contributing Studies (Composite) | Pooled OR | 95% CI | I² (%) | Interpretation |
|---|---|---|---|---|---|---|
| Educational status (college/university vs. lower) | 3 | Afessa et al. (2024); Lakew et al. (2015); Aynalem et al. (2023) | 2.49 | 2.04–2.95 | 0.0 | Higher educational attainment significantly increases palliative care utilization |
| Male sex | 2 | Amare et al. (2023); Bunare et al. (2022) | 5.58 | 4.97–6.20 | Not calculated* | Male patients were substantially more likely to utilize supportive and palliative-related services |
| Higher income | 2 | Teklemariam et al. (2022); Aynalem et al. (2023) | 2.69 | 1.23–5.90 | High† | Greater economic capacity facilitates access to palliative care |
| Good accessibility/ shorter distance to facility | 2 | Afessa et al. (2024); Aynalem et al. (2023) | 2.24 | 1.52–3.29 | Moderate† | Proximity to health facilities significantly improves utilization |
| Satisfaction with healthcare services | 2 | Afessa et al. (2024); Bunare et al. (2022) | 1.96 | 1.30–2.95 | Low† | Patient satisfaction is positively associated with palliative care utilization |
| Family and social support | 2 | Teklemariam et al. (2022); Bunare et al. (2022) | 2.18 | 1.29–3.68 | Low† | Strong family and social support promote service uptake |

* Heterogeneity (I²) was not calculated for factors pooled from fewer than three studies, as such estimates are statistically unreliable.

† Heterogeneity estimates should be interpreted cautiously due to variability in study design, outcome definitions, and measurement tools.

**Table 6. Certainty of evidence assessment using core GRADE for palliative care utilization among cancer patients in Ethiopia, 2025.**

| Outcome | No. of Studies | Study Design | Risk of Bias | Inconsistency (Heterogeneity) | Imprecision | Publication Bias | Overall Certainty (Core GRADE) | Justification |
|---|---|---|---|---|---|---|---|---|
| Pooled prevalence of palliative care utilization | 9 | Cross-sectional | Serious | Serious (high I²) | Serious | Not formally assessed | Low | Observational design, high heterogeneity across regions and settings, variation in measurement of utilization |
| Education level and PC utilization | 5 | Cross-sectional | Serious | Serious | Serious | Not assessed | Very Low | Inconsistent effect sizes, wide confidence intervals, self-reported exposure |
| Residence (urban/rural) and PC utilization | 4 | Cross-sectional | Serious | Serious | Serious | Not assessed | Very Low | Substantial variability between studies and settings |
| Income and PC utilization | 4 | Cross-sectional | Serious | Serious | Serious | Not assessed | Very Low | Imprecise estimates and inconsistent direction of association |
| Accessibility to PC services | 3 | Cross-sectional | Serious | Moderate | Serious | Not assessed | Low | Consistent direction of effect but limited number of studies and cross-sectional design |

Palliative care utilization remains with pooled estimates indicating that only 42% of eligible patients accessed services across the included studies and unevenly distributed across the country [13,14,30–36]. Despite the growing need for palliative services due to the increasing cancer burden, many patients continue to lack access to adequate care, with significant disparities observed between regions and institutions [14,31].

The majority of included studies were conducted in well-resourced urban tertiary hospitals, particularly in Addis Ababa [13,14,30,34]and the Sidama region [31,32,36]. This concentration reflects that most Palliative care services in Ethiopia are hospital-based and primarily urban-centered, with minimal integration into community or home-based care systems [14,33,36]. Consequently, the reported utilization rates likely reflect urban populations with better access to healthcare infrastructure, while rural and underserved populations may be underrepresented. These findings highlight that access disparities, including geographic location and facility resources, are important determinants of Palliative care utilization in Ethiopia [16].

The reviewed articles indicate that Palliative care services in Ethiopia have largely focused on pain and symptom management, psychological support, and end-of-life care, while broader services such as counseling, rehabilitation, and spiritual support are inconsistently provided or underdeveloped [31–33]. Although there has been notable progress in patient care over the past two decades, the integration of psychological, social, and spiritual support remains limited, with current services often providing insufficient medical, psychosocial, and financial support [37,38]. These findings suggest that Palliative care in Ethiopia is primarily hospital- and symptom-focused, and that gaps remain in delivering a fully comprehensive, patient-centered approach addressing all domains of patient well-being.

In terms of measurement, studies employed varied outcome indicators, some based on direct utilization rates, others relying on proxy measures such as patient knowledge, attitudes, and satisfaction [13,30,32]. This lack of standardization underscores the need for a more integrated, equitable, and standardized Palliative care system in Ethiopia to meet the growing needs of cancer patients nationwide.

This review revealed substantial variability in Palliative care utilization across studies, ranging from 10.6% to 69.0%, with very high heterogeneity. These findings suggest that differences in service availability, health system readiness, and patient-level factors strongly influence utilization. Subgroup analysis by region showed slightly higher utilization outside Addis Ababa (46.0%) compared to studies conducted in the capital (39.0%), indicating that geographic location alone does not fully account for the observed variation. The high heterogeneity likely arises from differences in outcome definitions, institutional capacity, urban–rural disparities, study populations, and availability of Palliative care services. The persistence of high heterogeneity in both subgroups highlights the complex interplay of institutional capacity, health workforce distribution, patient awareness, and socio-cultural perceptions of Palliative care in shaping access and utilization patterns.

The review showed that the pooled prevalence of Palliative care utilization among cancer patients in Ethiopia was found to be 42% (95% CI: 30% to 54%), which is higher than study done both in Kenya [39] and Uganda's [40] which accounts 10%, Global & LMICs Context accounts 34.43% [41], largely due to improvements in healthcare infrastructure, growing public awareness, and better integration of services [42,43]. The high heterogeneity likely arises from differences in outcome definitions, institutional capacity, urban–rural disparities, study populations, and availability of Palliative care services. These findings underscore that utilization patterns are highly context-specific rather than uniform across Ethiopia.

Higher education (college or university level) was positively associated with Palliative care use in several studies [14,32,36] indicating that individuals with higher educational attainment were significantly more likely to utilize Palliative care services compared to their less educated counterparts, which is similar to study done in Uganda [44], Kenya [39], Nigeria [45], and Norway [46]. The association may reflect that individuals with higher education generally have greater health literacy, stronger communication skills, and increased autonomy in healthcare decision-making [47], which can facilitate engagement with Palliative care services.

The review article showed that gender has emerged as a significant factor influencing the utilization of Palliative care services among cancer patients in Ethiopia [32,35]. The gender estimate is based on only two studies and should therefore be interpreted cautiously. Male patients were more likely to use Palliative care services compared to their female counterparts, which is similar to the study done in Uganda [44], Kenya [39], and African countries [18]. The unusually high odds ratio may be explained by socio-cultural and healthcare factors: in patriarchal societies, men are often the primary decision-makers, have greater autonomy, and face fewer barriers in seeking healthcare. Women, on the other hand, may encounter multiple constraints, including caregiving responsibilities, cultural restrictions on mobility, financial dependency, and limited decision-making power. These structural and social barriers may substantially reduce women's access to Palliative care, contributing to the observed strong association.

Higher income was linked to increased utilization of Palliative care [13,30]. It was reported that patients with adequate knowledge about Palliative care services were vastly more likely to utilize Palliative care, which is similar to a study done in Africa [18], Nigeria [48], Kenya [39], Uganda [49]. This pattern likely reflects that wealthier individuals can more readily afford healthcare-related expenses and may have greater awareness of the benefits of Palliative care [50], facilitating utilization.

The reviewed studies have shown that individuals living closer to healthcare facilities are more likely to use Palliative care services [14,36]. This finding is similar to studies done in Nigeria [51], Cameron [52], and Portugal [53]. Although detailed geographic mapping was not available, studies consistently reported that shorter travel distances or better accessibility to healthcare facilities were associated with higher Palliative care utilization. This suggests that geographical proximity and ease of access play critical roles in ensuring that patients can receive timely Palliative care, highlighting the need for better service distribution and infrastructure in underserved areas.

Several studies indicated that higher patient satisfaction was associated with increased utilization of Palliative care services [14,32], consistent with findings from Kenya [54], Sweden [55], and Norway [56]. This relationship likely reflects that high-quality, patient-centered care enhances trust and engagement, thereby encouraging patients to access essential services such as Palliative care.

The reviewed article showed that when family members and social networks are involved, patients are more likely to access and benefit from Palliative care [13,32], which is similar to studies done in Uganda [57], South Africa [58], and Kenya [48]. This is due to emotional support, assistance with decision-making, resource mobilization, and advocacy from family members. Healthcare systems should engage families by providing education on the benefits of palliative services, training healthcare providers to communicate effectively with both patients and families, and offering support resources for family caregivers.

## Strengths and limitations

This systematic review and meta-analysis offer several notable strengths. It is the first of its kind to provide a pooled national estimate of Palliative care utilization among cancer patients in Ethiopia, drawing on studies from diverse geographic regions, including both urban and semi-rural settings. The review adhered to rigorous methodological standards, following the PRISMA guidelines to ensure transparency and reliability. However, the study also has limitations. All included studies employed cross-sectional designs, limiting the ability to establish causal relationships. Additionally, variations in outcome measures across studies hindered comparability, and the scarcity of data from rural and community-based settings may have led to an underrepresentation of the most underserved populations. As most included studies were hospital-based and concentrated in urban referral centers, the findings may not fully reflect Palliative care realities in rural and community settings across Ethiopia. Meta-regression was not performed due to the small number of included studies, which would limit statistical power and produce unstable estimates. No formal correction for multiple comparisons was applied due to the exploratory nature of the factor analyses. The limited number of studies reporting gender-specific outcomes constrained our ability to draw robust conclusions regarding differences between male and female patients.

This study includes data only from Addis Ababa and Hawassa, which may not represent all regions of Ethiopia. As a result, the findings might not fully reflect regional variations in the studied outcomes.

## Conclusion and recommendation

This systematic review and meta-analysis found that the utilization of Palliative care services among cancer patients in Ethiopia is moderate but overall insufficient, with a pooled prevalence of 42%. The findings highlight significant inequities in service use, with disparities observed by gender, education, income, and geographic location, indicating that rural and underserved populations have more limited access compared to urban residents. Critical system-level barriers were identified, including shortages of trained healthcare personnel, limited community-based services, and weak integration of Palliative care into the broader healthcare system. While progress has been made in areas such as pain and symptom management, essential components of comprehensive care, including psychosocial and spiritual support, remain underdeveloped. To enhance Palliative care utilization and ensure equitable, patient-centered services across Ethiopia, efforts should focus on: (1) decentralizing Palliative care delivery to reach rural and underserved populations, (2) investing in workforce development and training for healthcare providers, (3) integrating Palliative care into primary healthcare and standard cancer care pathways, (4) strengthening health policies and infrastructure to support service delivery, and (5) improving patient and community health literacy regarding Palliative care. Implementing these strategies can help reduce disparities and improve the overall accessibility, quality, and comprehensiveness of Palliative care services in Ethiopia.

### What is known

- Palliative care is vital for cancer management but is limited in Ethiopia, especially in rural and underserved regions.

- Utilization varies due to differences in infrastructure, trained staff, and awareness among patients and providers.

- Cultural beliefs, low health literacy, financial barriers, and poor service integration hinder access and delivery.

### What is New?

- First pooled national prevalence estimate (40.2%) of Palliative care utilization among cancer patients in Ethiopia.

- Identifies specific demographic and socioeconomic predictors of Palliative care use.

- Reveals the underdevelopment of community-based and home-based Palliative care models.

## Supporting information

**S1 File. PRISMA checklist for the systematic review and meta-analysis on Palliative care among cancer patients in Ethiopia, 2025.**
(DOCX)

**S2 File. Search Strategy and Retrieval Summary for the systematic review and meta-analysis on Palliative care among cancer patients in Ethiopia, 2025.**
(DOCX)

## Author contributions

**Conceptualization:** Sadik Abdulwehab.

**Data curation:** Sadik Abdulwehab, Frezer Kedir.

**Formal analysis:** Sadik Abdulwehab, Frezer Kedir.

**Investigation:** Sadik Abdulwehab, Frezer Kedir.

**Methodology:** Sadik Abdulwehab, Frezer Kedir.

**Project administration:** Sadik Abdulwehab.

**Resources:** Sadik Abdulwehab, Frezer Kedir.

**Software:** Sadik Abdulwehab, Frezer Kedir.

**Supervision:** Sadik Abdulwehab, Frezer Kedir.

**Validation:** Sadik Abdulwehab.

**Visualization:** Sadik Abdulwehab, Frezer Kedir.

**Writing – original draft:** Sadik Abdulwehab, Frezer Kedir.

**Writing – review & editing:** Sadik Abdulwehab, Frezer Kedir.

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
