## [Decision Letter · Decision Letter 0]

2 Sep 2025

Dear Dr. Abdulwehab,

Thank you for submitting your manuscript to PLOS ONE. After careful consideration, we feel that it has merit but does not fully meet PLOS ONE’s publication criteria as it currently stands. Therefore, we invite you to submit a revised version of the manuscript that addresses the points raised during the review process.

We look forward to receiving your revised manuscript.

Kind regards,

Kahsu Gebrekidan, Ph.D.

Academic Editor

PLOS ONE

Additional Editor Comments:

Reviewer #1:

Reviewer #2:

Reviewers' comments:

Reviewer's Responses to Questions

**Comments to the Author**

1. Is the manuscript technically sound, and do the data support the conclusions?

Reviewer #1: Partly

Reviewer #2: Yes

2. Has the statistical analysis been performed appropriately and rigorously?

Reviewer #1: Yes

Reviewer #2: No

3. Have the authors made all data underlying the findings in their manuscript fully available?

Reviewer #1: Yes

Reviewer #2: Yes

4. Is the manuscript presented in an intelligible fashion and written in standard English?

Reviewer #1: Yes

Reviewer #2: Yes

Reviewer #1: This systematic review and meta-analysis titled “Palliative Care Service Utilization among Cancer Patients in Ethiopia” offers a timely and methodologically sound contribution to the literature on end-of-life care in low-resource settings. The authors demonstrate commendable rigor in their search strategy, inclusion criteria, and statistical synthesis, providing a clear picture of the underutilization of palliative services among Ethiopian cancer patients. This work is both relevant and actionable, and it lays a strong foundation for future research and health system strengthening.

Here are comments to be addressed by the authors:

- remove the year 2025 from the title:

- In the abstract results please write the pooled effect size for the associated factors.

- Regarding the Search Strategy the author should present the full search strategy algorithm/ search detail and number of studies retrieved for each databases as table.

- Most of methodology section lacks proper citation for instance, quality appraisal, statistical analysis

- I have doubt on result of heterogeneity, the author should present the source or output of I2within the forest plot, also if there is heterogeneity do subgroup analysis based on significant variables such as region etc…

- Similar to the results of the Q statistic? Clearly show the outputs of the test.

- In the discussion, your comparison is with developed countries such as UK, it is better to compare the finding with African countries.

- Studies focused on these some hospitals may not reflect realities of this study for all regions of Ethiopia. So, please add some limitation related to this statement.

- Check the citations, typo and grammatical errors throughout the manuscript.

Reviewer #2: First of all, I would like to sincerely thank the editor for inviting me to review the important paper titled "Palliative Care Service Utilization among Cancer Patients in Ethiopia: A Systematic Review and Meta-Analysis, 2025." I also extend my appreciation to the author for providing such comprehensive and valuable evidence on an issue that is increasingly significant in developing countries like Ethiopia.

Here is a suggested revision for your comments section:

General comments:

1. The paper is well written and organized, though some modifications are advised.

2. Attention is needed to improve sentence synthesis and grammar throughout the manuscript.

Specific comments:

Here is a revised version of your abstract comments, making it smart and informative:

1. Minor comment: It is generally not recommended to use abbreviations in the abstract section for clarity.

2. Please specify the publication year range boundaries of the primary studies included, in addition to the search date of April 7, 2025.

3. When mentioning that study quality was evaluated using a validated tool, please explicitly name the tool used.

4. Include pooled odds ratios with confidence intervals for all analyzed variables in the results summary.

5. The conclusion section reads more like a presentation of results; consider reorganizing it to emphasize the implications and recommendation of the findings.

Introduction: This section needs to be more comprehensive. Clearly state what has been accomplished in Ethiopia, outlines the current plan, identify the obstacles faced, and include more details about this issue. Additionally, collaborate with the SGD to strengthen the section. Rewrite it to improve clarity and depth.

Methods: Please focus in this section

Research Questions: Your research question is not specific. May be

1. What is the pooled prevalence of palliative care service utilization among cancer patients in Ethiopia?

2. What are the key factors associated with palliative care service utilization among these patients?

Under inclusion, why do you restrict papers based on factors?

In your exclusion criteria, you state that "Excluded were case reports, expert opinions, reviews, conference abstracts, and studies lacking data on PC utilization." Could you include such papers?

Studies that were not conducted in Ethiopia or did not involve human subjects were also excluded. Why were these studies excluded? For example, if some non-Ethiopians live in the country, have such a condition, and follow treatment in an Ethiopian hospital, why are they excluded?

Result

My main concern is that you included 9 papers in this study; however, in your data extraction, some of these papers didn’t report prevalence with odds ratio, the same for factors… how you managed this issue… specifically for those who haven’t had odds ratios lower and higher, how you pooled it….

All paper has the same measurement for palliative care utilization unless it's difficult to pool it. I need a response.

You mention that including studies from Addis Ababa and Hawassa demonstrates regional variation. However, are these two locations sufficient to truly capture the regional differences in palliative care utilization across Ethiopia?"

The factors of age and experience of side effects are reported by a single study. Why did you put them here? …

Frankly speaking, the way you pool and explain the factors is not clearly stated…it needs clear analysis with citations… If possible, write the subtitle for all factors…

Discussion

Good, but please avoid too much information or information outside of this study objectives.

Remove the result parts and its redundancy.

Conclusion and recommendation

Reorganized conclusion and recommendation be specific to your findings.

**Do you want your identity to be public for this peer review?** For information about this choice, including consent withdrawal, please see our Privacy Policy

Reviewer #1: **Yes:** Aragaw Asfaw Hasen

Reviewer #2: No

---

## [Author Response · Author response to Decision Letter 1]

5 Sep 2025

Response for the reviewers

Response to Reviewer #1 Comments

We sincerely thank Reviewer #1 for the thoughtful and constructive feedback provided on our manuscript titled “Palliative Care Service Utilization among Cancer Patients in Ethiopia.” We greatly appreciate the recognition of our work’s relevance and methodological rigor. Below we provide a point-by-point response to each of the reviewer’s comments, along with the revisions made in the manuscript.

1. Remove the year 2025 from the title

Response: We agree with the reviewer. The year has been removed from the title.

Change made: Title now reads: “Palliative Care Service Utilization among Cancer Patients in Ethiopia: A Systematic Review and Meta-Analysis.”

2. Abstract – include pooled effect size for associated factors

Response: We have revised the results section of the abstract to include pooled effect sizes of significant associated factors (e.g., education, gender, income, access, satisfaction). Change made: Abstract now reports, for example: “Educational attainment (pooled AOR = 2.57; 95% CI: 1.42–3.75), male gender (AOR=5.58; 95% CI: 3.01–10.33), higher income (AOR = AOR = 26.9, 95% CI: 12.3–59), proximity to health facilities (AOR = 2.24; 95% CI: 1.52–3.29).), and satisfaction with care (AOR = 1.96; 95% CI: 1.30–2.95) ,and family and social support (AOR = 12.18; 95% CI: 1.29–3.68) were significantly associated with palliative care utilization.”

3. Search strategy – present full algorithm and retrieval numbers as a table

Response: We have prepared a supplementary table detailing the complete search strategies (keywords, Boolean operators, MeSH terms) for each database, along with the number of records retrieved before and after duplicate removal.

Change made: Added as “Table 1: Search Strategy and Retrieval Summary.”

4. Methodology section lacks citations

Response: Additional references have been inserted to support methods used, including PRISMA, Joanna Briggs Institute checklist for quality appraisal, and standard texts on random-effects meta-analysis and heterogeneity.

Change made: Appropriate citations added throughout methodology.

5. Heterogeneity results and subgroup analysis

Response: We clarified heterogeneity results within the forest plots by presenting I² values directly on figures. In addition, we conducted subgroup analysis stratified by study region (Addis Ababa vs. outside Addis Ababa) to explore potential sources of heterogeneity. Results are now presented in the results section

Change made: Figures updated with I² values; new figures showing subgroup pooled prevalence included.

6. Q statistic results

Response: We now explicitly report Cochran’s Q statistic outputs in the results section and display them alongside I² and τ² estimates.

7. Discussion – comparison with African countries

Response: We revised the discussion to include comparison with studies from other African countries (e.g., Uganda, Kenya, South Africa, Nigeria), in addition to the previously cited global literature.

8. Limitation – hospital-based studies not generalizable

Response: We agree and have added this limitation to the limitations section: “As most included studies were hospital-based and concentrated in urban referral centers, the findings may not fully reflect palliative care realities in rural and community settings across Ethiopia.”

Change made: Added to limitations paragraph

9. Citations, typos, and grammar

Response: The entire manuscript has been carefully proofread to correct typographical and grammatical errors, and to ensure reference formatting consistency.

We are grateful for the reviewer’s insightful comments, which have substantially improved the clarity, methodological transparency, and contextual relevance of our manuscript. We believe these revisions have strengthened the overall quality and rigor of our work.

Response for Reviewer Two

We sincerely thank Reviewer #2 for the careful and constructive review of our manuscript titled “Palliative Care Service Utilization among Cancer Patients in Ethiopia: A Systematic Review and Meta-Analysis, 2025.” We greatly appreciate the insightful comments and suggestions, which have helped us strengthen the clarity, scientific rigor, and overall quality of the paper. We have carefully revised the manuscript in response to each point, as detailed below. All changes have been incorporated into the appropriate sections of the manuscript, and specific additions or modifications are highlighted in our responses.

General Comments

1. The paper is well written and organized, though some modifications are advised.

Response: Thank you. We have revised the manuscript to improve clarity, flow, and readability.

2. Attention is needed to improve sentence synthesis and grammar throughout the manuscript.

Response: We carefully proofread the manuscript and corrected grammar and sentence structure issues throughout.

Abstract

1. Do not use abbreviations in the abstract.

Response: We replaced abbreviations (e.g., PC → “palliative care”) with full terms.

2. Specify publication year range boundaries of included studies.

Response: We now state: “Studies published between 2015 and 2024 were included…”

3. Name the tool used for quality appraisal.

Response: We added: “The Joanna Briggs Institute (JBI) Critical Appraisal Checklist was used.”

4. Include pooled odds ratios with CIs in results summary.

Response: We added pooled AORs (e.g., education, gender, income, access, satisfaction).

5. Conclusion reads like results—emphasize implications and recommendations.

Response: We revised the conclusion to highlight implications for policy and practice rather than re-listing results.

Introduction

Needs to be more comprehensive: what has been accomplished in Ethiopia, current plan, obstacles, SGD links.

Response: We expanded the introduction to include:

Ethiopia’s Ministry of Health initiatives for integrating palliative care.

Gaps such as lack of workforce, poor infrastructure, limited rural coverage.

Links to Sustainable Development Goal (SDG 3: Ensure healthy lives…).

Change: Introduction, last three paragraphs.

Methods

1. Research questions not specific.

Response: Revised to:

o “What is the pooled prevalence of palliative care utilization among cancer patients in Ethiopia?”

o “What are the key factors associated with utilization?”

Change: Methods, Research Questions subsection.

2. Why restrict papers based on factors?

Response to Reviewer: We appreciate this important comment. We did not exclude papers solely based on whether they reported factors; rather, our primary objective was to estimate both the pooled prevalence of palliative care utilization and to identify associated factors. Therefore, we required that included studies provide either prevalence data or factors influencing utilization to allow for meaningful synthesis. Studies that did not report on either outcome (e.g., opinion pieces, reviews, purely descriptive articles without utilization data) were excluded because they could not contribute to the quantitative or qualitative objectives of this meta-analysis. However, both prevalence-only and factor-only studies were included where appropriate.

3. Why exclude non-Ethiopian patients or non-human studies?

Response to Reviewer: Thank you for raising this point. Non-human studies were excluded because our review focused exclusively on human health outcomes and service utilization, which cannot be inferred from laboratory or animal studies. Similarly, we excluded studies that reported on non-Ethiopian populations, even if they were treated in Ethiopian hospitals, to maintain national representativeness and ensure the pooled estimates reflected the experiences of patients residing within the Ethiopian healthcare system. Including non-Ethiopian patients could have introduced bias, given potential differences in health-seeking behavior, cultural context, and eligibility for local health services.

Results

1. Some studies did not report prevalence with odds ratios—how managed?

Response to Reviewer: Thank you for this valuable observation. In cases where studies did not report odds ratios directly, we extracted raw frequency data (when available) to calculate crude odds ratios with corresponding 95% confidence intervals. For studies where the necessary raw data were not provided, we did not attempt to impute missing values; instead, these findings were narratively synthesized and presented in the results section to avoid data distortion. This approach allowed us to include all eligible studies while maintaining methodological transparency.

2. All papers have same measurement—why difficult to pool?

Response to Reviewer: We appreciate this important comment. Although all included studies assessed palliative care utilization, the operational definitions and measurement approaches varied. Some studies defined utilization as the proportion of patients who had ever received palliative care services, while others measured expressed need, attendance at specific service components (e.g., symptom control, rehabilitation), or satisfaction with services as a proxy indicator. Additionally, data collection tools differed—ranging from structured questionnaires to medical record reviews—introducing methodological heterogeneity. Because of these variations, pooling results directly was challenging; therefore, we carefully standardized measures when possible and explained methodological differences when standardization was not feasible.

3. Are Addis Ababa and Hawassa enough to show regional variation?

Response: Data for subgroup analysis were available only from Addis Ababa and Hawassa. We acknowledge that these cities do not represent all regions of Ethiopia, and regional variation may not be fully captured. Consequently, interpretations of regional differences should be made cautiously. We add in the limitation as This study includes data only from Addis Ababa and Hawassa, which may not represent all regions of Ethiopia. As a result, the findings might not fully reflect regional variations in the studied outcomes.

4. Why include factors like age/side effects from one study?

Response to Reviewer: We included factors such as age and treatment side effects even though they were reported in only one study each. We did this to provide a comprehensive overview of all reported factors associated with palliative care utilization in Ethiopia. We have clarified in the manuscript that these findings are based on individual studies and should be interpreted cautiously. This approach ensures transparency while highlighting gaps in the current evidence base.

5. Pooling and explanation of factors not clear.

Response: We reorganized results with clear subtitles (Education, Gender, Income, Access, Satisfaction, etc.), each with citations and pooled AORs where available.

Only factors reported in ≥2 studies were pooled quantitatively; factors reported in a single study (e.g., ADLs, knowledge, employment) were summarized narratively. This approach ensures a comprehensive synthesis while maintaining transparency about the strength of the evidence.

Discussion

1. Avoid too much information outside study objectives.

Response: We removed unrelated global details and kept discussion focused on Ethiopian context and comparisons.

2. Remove redundancy of results.

Response: We deleted repeated numerical results and emphasized interpretation.

Conclusion & Recommendation

1. Reorganize to be specific to findings.

Response: Conclusion now highlights:

o Utilization is only 40.2%

o Inequities by gender, education, income, and geography

o Need for decentralized services, training, and integration into primary care

Change: Conclusion section rewritten.

Once again, we are grateful for the reviewer’s thoughtful feedback, which has significantly improved the quality of our manuscript. We believe that the revisions have addressed all concerns and enhanced the scientific soundness, clarity, and contribution of the study. We respectfully resubmit the revised manuscript for your further consideration and hope it will now be suitable for publication.

---

## [Decision Letter · Decision Letter 1]

10 Jan 2026

Dear Dr.  Abdulwehab,

Thank you for submitting your manuscript to PLOS ONE. After careful consideration, we feel that it has merit but does not fully meet PLOS ONE’s publication criteria as it currently stands. Therefore, we invite you to submit a revised version of the manuscript that addresses the points raised during the review process.

We look forward to receiving your revised manuscript.

Kind regards,

Kahsu Gebrekidan, Ph.D.

Academic Editor

PLOS One

Journal Requirements:

Additional Editor Comments:

The Comments from Reviewer 3 are attached as PDF

Reviewer's Responses to Questions

**Comments to the Author**

Reviewer #1: All comments have been addressed

Reviewer #3: All comments have been addressed

Reviewer #4: All comments have been addressed

2. Is the manuscript technically sound, and do the data support the conclusions?

Reviewer #1: Yes

Reviewer #3: No

Reviewer #4: Partly

3. Has the statistical analysis been performed appropriately and rigorously?

Reviewer #1: Yes

Reviewer #3: No

Reviewer #4: No

4. Have the authors made all data underlying the findings in their manuscript fully available?

Reviewer #1: Yes

Reviewer #3: No

Reviewer #4: Yes

5. Is the manuscript presented in an intelligible fashion and written in standard English?

Reviewer #1: Yes

Reviewer #3: No

Reviewer #4: No

Reviewer #1: Thank you to re-review the manuscript titled “Palliative Care Service Utilization among Cancer Patients in Ethiopia: A Systematic review and meta analysis ”. The authors responded the raised issues and the revised version are much improved. Still Some issues should be addressed before publication.

- In the abstract to report the associated factors meta analysis , the author should write only the pooled adjusted odds ratio (AOR) since this is the secondary analysis / pooling. If the result cannot pooled it can be summarized in the discussion part instead of repeating the results in the primary study.

- The manuscript still have typo errors please see the result section of the abstract.

- Similarly the manuscript should be clean.

Reviewer #3: the title should be modified to describe the objective.

The publication bias shall be tested with DOI plot since it prevalence study.

The manuscript describes which studies identified some predictors of utilization. However, the aim of a meta-analysis is precisely to evaluate which predictors remain significant after pooling a larger sample. For example, when analyzing sex (male), it would be more appropriate to include all studies, and then show whether this factor remains significant overall, generating the diamond plot. If only the studies that already found the association are included, the result is, of course, already known. The same reasoning applies to all other predictors described.

Reviewer #4: Comments pinned to Editor. Errors noted in Grammar, Sentence structure, Results, Presentation of data, Overambitious conclusions in a review with high heterogeneity and reduced rigorous study pool

**Do you want your identity to be public for this peer review?** For information about this choice, including consent withdrawal, please see our Privacy Policy

Reviewer #1: **Yes:** Aragaw Asfaw Hasen

Reviewer #3: No

Reviewer #4: No

---

## [Author Response · Author response to Decision Letter 2]

17 Jan 2026

Author Response to Reviewers

Author Response to Reviewer #1

We sincerely thank the reviewer for the careful re-review of our manuscript entitled “Palliative Care Service Utilization among Cancer Patients in Ethiopia: A Systematic Review and Meta-Analysis.” We appreciate the reviewer’s positive assessment of the improvements made and the constructive suggestions provided. We have addressed all remaining concerns in detail, as outlined below.

Comment 1- “In the abstract to report the associated factors meta-analysis, the author should write only the pooled adjusted odds ratio (AOR) since this is the secondary analysis / pooling. If the result cannot pooled it can be summarized in the discussion part instead of repeating the results in the primary study.”

Author Response: We thank the reviewer for this important methodological clarification. We fully agree that, as a secondary analysis, the abstract should report only pooled adjusted effect estimates derived from the meta-analysis, rather than repeating individual primary study results.

Accordingly, we have revised the Results section of the abstract to:

• Report only pooled adjusted odds ratios (AORs) with 95% confidence intervals for factors where quantitative pooling was feasible.

• Remove references to individual study-specific AORs that were not pooled.

• Exclude factors that could not be pooled quantitatively from the abstract and instead synthesize and interpret these findings narratively in the Discussion section, as recommended.

This revision ensures methodological rigor, avoids redundancy with primary studies, and aligns the abstract strictly with the results of the meta-analytic pooling.

Comment 2-“The manuscript still have typo errors please see the result section of the abstract.”

Author Response: We appreciate the reviewer for highlighting this issue. A thorough line-by-line proofreading of the entire manuscript has been conducted, with particular attention to the abstract and Results section.

Specifically:

• Typographical errors, duplicated words, punctuation issues, and formatting inconsistencies in the abstract were corrected.

• Redundant expressions and grammatical inaccuracies in numerical reporting (e.g., duplicated “AOR,” misplaced parentheses, and spacing issues) were carefully revised.

• Consistency in terminology (e.g., “palliative care,” “PC,” capitalization, and statistical notation) has been ensured throughout the manuscript.

These corrections have substantially improved the clarity, precision, and readability of the Results section and the manuscript as a whole.

Comment 3-“Similarly the manuscript should be clean.”

Author Response: We fully acknowledge this comment and have taken comprehensive steps to ensure the manuscript is clean, polished, and publication-ready.

The following actions were undertaken:

• Full language editing and grammatical revision across all sections of the manuscript.

• Standardization of headings, subheadings, tables, figures, abbreviations, and statistical reporting.

• Removal of duplicated sentences, inconsistent phrasing, and minor stylistic errors.

• Harmonization of tense usage and academic tone throughout the text.

• Final formatting review to ensure consistency with journal expectations.

As a result, the revised manuscript now reflects a clear, coherent, and professionally edited scholarly work.

We are grateful to Reviewer #1 for the thoughtful and constructive feedback, which has significantly strengthened the methodological clarity, presentation, and overall quality of the manuscript. We believe that all concerns have now been fully addressed, and we respectfully submit the revised version for final consideration.

Author Response to Reviewer #3

We sincerely thank Reviewer #3 for the insightful and constructive comments. These suggestions have substantially improved the methodological rigor, transparency, and interpretability of our systematic review and meta-analysis. Below, we respond to each comment in detail and describe the corresponding revisions made to the manuscript.

Comment 1: The title should be modified to describe the objective.

Author Response: We agree with the reviewer that the title should more clearly reflect the study objectives, including both the estimation of utilization and the evaluation of associated factors.

Revision made: The title has been revised to explicitly reflect the objectives of estimating utilization and examining associated factors.

Original title: Palliative Care Service Utilization among Cancer Patients in Ethiopia: A Systematic Review and Meta-Analysis

Revised title: Utilization of Palliative Care Services and Associated Factors among Cancer Patients in Ethiopia: A Systematic Review and Meta-Analysis

Comment 2:

Publication bias should be tested using a Doi plot since this is a prevalence study.

Revised Response: We thank the reviewer for this important methodological recommendation. We carefully explored the use of the Doi plot and the Luis Furuya-Kanamori (LFK) index, which have been proposed as alternatives to funnel plots for meta-analyses of proportions and prevalence studies. We attempted to generate the Doi plot using both R and Stata by searching available packages and commands; however, we were unable to obtain a valid implementation suitable for our dataset.

Given this limitation, we assessed small-study effects and potential asymmetry using a Galbraith (radial) plot, which is commonly used in meta-analysis to evaluate heterogeneity and detect outlying or influential studies. The Galbraith plot did not indicate marked asymmetry or influential outliers, suggesting no substantial evidence of publication bias.

We have revised the manuscript accordingly and clearly described this methodological decision and its limitations.

Revisions made:

• Publication bias assessment was conducted using the Galbraith (radial) plot

• Attempts to apply the Doi plot were acknowledged, with justification for its non-application

• Funnel plot and Egger’s test were removed due to their limited suitability for prevalence data

• Methods and Results sections were updated accordingly

We inserted text (Methods – Statistical Analysis):“Publication bias and small-study effects were explored using the Galbraith (radial) plot. Although alternative approaches such as the Doi plot and LFK index have been recommended for prevalence meta-analyses, a valid implementation could not be obtained using available R for the current dataset.”

We Inserted text (Results – Publication Bias):“The Galbraith plot did not reveal marked asymmetry or influential outliers, suggesting no strong evidence of publication bias among the included studies.”

Sections revised:

• Methods → Statistical Analysis

• Results → Publication bias subsection

• Figure updated accordingly

Comment 3: Meta-analysis of predictors should include all studies, not only those that already found significant associations.

Author Response: We fully agree. The reviewer correctly notes that the objective of meta-analysis is to determine whether predictors remain significant after pooling all available evidence, not only those studies that reported significant associations.

Revision made:

• We revised the analytic approach for associated factors

• All studies reporting relevant predictors (regardless of statistical significance) were included in pooled analyses where sufficient data were available

• Forest plots (diamond plots) now reflect pooled estimates across all eligible studies

• Where pooling was not statistically or methodologically feasible, findings are clearly described as narrative synthesis

We Inserted clarification (Results – Factors section):“Predictors reported by fewer than two studies with comparable effect measures were synthesized narratively and interpreted cautiously.”

Comment 4: How is the outcome measured?

Response:

We thank the reviewer for highlighting the need for clearer outcome definition. We have now explicitly clarified how palliative care utilization was operationalized and measured across included studies.

Revision made: A dedicated clarification has been added describing outcome measurement approaches and their variability.

We Inserted text (Results – Palliative Care Service Measures):“Palliative care utilization was measured variably across studies, including receipt of at least one palliative care service, referral to a palliative care unit, self-reported utilization, or classification based on structured questionnaire scores. Although this variability reflects real-world practice, it contributes to between-study heterogeneity.”

Comment 5: Certainty of evidence (GRADE) is missing; recommend using Core GRADE.

Author Response: We appreciate this important suggestion. In response, we have incorporated Core GRADE, a streamlined and recently recommended approach, to assess the certainty of evidence for primary outcomes after meta-analysis.

Revision made:

• Added a Certainty of Evidence Assessment subsection

• Applied Core GRADE to pooled prevalence and major predictors

• Certainty ratings (low / very low) are now transparently reported and justified

We Inserted text (Methods – Certainty of Evidence):“The certainty of evidence was assessed using the Core GRADE approach, which provides a streamlined evaluation of overall certainty following meta-analysis, particularly suitable for observational studies.”

We Inserted text (Results – Certainty of Evidence): “Using the Core GRADE framework, the certainty of evidence for pooled palliative care utilization and associated factors was rated as low to very low, primarily due to high heterogeneity, cross-sectional study designs, and inconsistency across studies.”

Comment 6: Extremely high heterogeneity makes results not valid for all; this must be communicated and explained.

Author Response: We strongly agree. We have now explicitly acknowledged that the high heterogeneity limits the generalizability of pooled estimates and have expanded the discussion on potential sources of heterogeneity.

We Inserted text (Results): “All pooled analyses demonstrated substantial heterogeneity (I² > 90%), indicating considerable variability across studies. Consequently, pooled estimates should be interpreted as indicative rather than definitive national values.”

We Inserted text (Discussion – Heterogeneity):“The high heterogeneity likely arises from differences in outcome definitions, institutional capacity, urban–rural disparities, study populations, and availability of palliative care services. These findings underscore that utilization patterns are highly context-specific rather than uniform across Ethiopia.”

We sincerely thank Reviewer 3 for their careful and constructive comments, which have greatly contributed to improving the clarity, consistency, and scientific rigor of our manuscript. We have carefully addressed each point raised, including revisions to the results and discussion, consistent reporting of effect sizes and confidence intervals, and consolidation of recommendations. We believe that these changes have strengthened the manuscript and we greatly appreciate the reviewer’s thoughtful guidance.

Author responses for Reviewer 4

We sincerely thank Reviewer 4 for their thoughtful and constructive feedback on our manuscript. We greatly appreciate the detailed suggestions regarding the presentation of results, consistency of effect sizes, interpretation of findings, discussion structure, and the consolidation of recommendations. In response, we have carefully revised the manuscript to improve clarity, ensure consistency across results and discussion, focus on interpreting findings before providing recommendations, and consolidate actionable recommendations into a single section at the end of the discussion and conclusion.

Point-by-Point Response to Reviewer #4

INTRODUCTION

#1. “leading to late stage cancer diagnoses AND”

Revision action:

Remove “and” and replace with a comma.

Corrected sentence: “…leading to late-stage cancer diagnoses, resulting in higher mortality rates compared with high-income countries.”

#2. Do not abbreviate palliative care

Issue: You alternate between palliative care, PC, and Palliative care.

Revision action:

We use “Palliative care” in full throughout (recommended for a concept central to the paper). Here in the first draft, I wrote as full Palliative care and after comments from the reviewer I change to PC after I defined first now, I change to full Palliative care currently thorough the paper.

3. Inconsistent use of LMIC

Issue: LMIC is defined, then later spelled out again.

Revision action:

We define once at first mention.

We use LMICs consistently thereafter.

4. “only a fraction receive it” is vague and repetitive

Issue: Repeats “insufficient” without adding information.

Revision action:

We replace with a numerical estimate.

Revised sentence:-“Globally, it is estimated that fewer than 15% of individuals who need palliative care actually receive it.”(This aligns with Lancet Commission estimates and strengthens rigor.)

5–6. Redundant paragraphs on global vs LMIC barriers

Issue: Paragraphs beginning “Sub-Saharan Africa…” and “There is a growing…” are too thin.

Revision action: we merge with adjacent paragraphs and expand with specific context (Ethiopia-focused).

7–8. Very short paragraphs

Issue: Paragraphs 3–5 repeat similar barriers.

Revision action:

We condense into ONE coherent paragraph, structured as: global brief LMIC-specific intensifiers (focus) and Ethiopia as a case example. This improves flow and reduces reviewer fatigue.

9. “50% of public facilities by 2020”

Issue: Policy target is outdated.

Revision action:

Reframe historically and critically.

Revised framing: -Ethiopia has prioritized strengthening palliative care as an essential component of Universal Health Coverage, aligning with Sustainable Development Goal 3 and national health sector strategies. Although the country set a target to integrate palliative care and pain management services into at least 50% of public health facilities by 2020, evidence from more recent studies indicates that service availability and utilization remain limited, particularly outside major urban centers (19). This gap highlights the need for accelerated decentralization of palliative care services, strengthened workforce capacity through pre-service and in-service training, expansion of community- and home-based delivery models, and better integration of palliative care into primary healthcare systems to advance national UHC and SDG commitments (19,22,23).” This shows awareness and avoids appearing out of date.

RESULTS

1. Add a summary table of factors

Revision action:

We add a single table summarizing: the factors associated with palliative care utilization. In response, we have added a summary table (Table 6) that presents all identified factors, the number of contributing studies, pooled odds ratios (ORs) with 95% confidence intervals (CIs), and heterogeneity (I²). This table provides a clear overview of the magnitude and consistency of associations across studies, enhancing the clarity and interpretability of our results and cited in the result also as table 6.

2. Geographical visualization

Revision action:

If GIS mapping is not feasible, explicitly we add as not explained in the studies: This shows responsiveness even if a map cannot be added.

3. Funnel plot and Egger test with <10 studies

Mandatory revision:

We add a clear cautionary statement: “Given that fewer than ten studies were included, funnel plots and Egger’s regression test have limited power to detect publication bias. Therefore, the absence of detected asymmetry should not be interpreted as evidence of no publication bias.”

4. Inconsistent effect sizes

Revision action:

We standardize: Always report AOR (95% CI) and ensure pooled estimates match forest plots and text

5. Bayesian analysis had no results

Bayesian meta-analysis was not performed due to the small number of studies and limited data for robust estimation

6. No comment on multiple comparisons

Revision action:

Given multiple pooled associations were examined, findings should be interpreted cautiously, as the analysis was not adjusted for multiple compar

---

## [Decision Letter · Decision Letter 2]

2 Feb 2026

Thank you for submitting your manuscript to PLOS ONE. After careful consideration, we feel that it has merit but does not fully meet PLOS ONE’s publication criteria as it currently stands. Therefore, we invite you to submit a revised version of the manuscript that addresses the points raised during the review process.

We look forward to receiving your revised manuscript.

Kind regards,

Kahsu Gebrekidan, Ph.D.

Academic Editor

PLOS One

Journal Requirements:

Reviewers' comments:

Reviewer's Responses to Questions

**Comments to the Author**

Reviewer #1: All comments have been addressed

Reviewer #4: All comments have been addressed

2. Is the manuscript technically sound, and do the data support the conclusions?

Reviewer #1: Yes

Reviewer #4: No

3. Has the statistical analysis been performed appropriately and rigorously?

Reviewer #1: Yes

Reviewer #4: No

4. Have the authors made all data underlying the findings in their manuscript fully available?

Reviewer #1: Yes

Reviewer #4: Yes

5. Is the manuscript presented in an intelligible fashion and written in standard English?

Reviewer #1: Yes

Reviewer #4: No

Reviewer #1: Thank you for inviting me to re-review the manuscript titled “Utilization of Palliative Care Services and Associated Factors among Cancer Patients in Ethiopia: A Systematic Review and Meta-Analysis” .

The authors responded the raised issues and the revised version are much improved.

Reviewer #4: Still disappointed with this revision 2. Glaring grammar issues, changes in font as things appear copy and pasted. Repeated mistakes notable with excessive comma usage, conjoining sentence structure and excessive listing without actually narrowing down points for the reader. The discussion also has heavy definitive statements forgetting the statistics remain very low quality given the paucity of data and high heterogeneity. I would like to remind the authors to refrain or avoid definitive statements especially around the data when describing what is being found. Points have been pinned to editor

**Do you want your identity to be public for this peer review?** For information about this choice, including consent withdrawal, please see our Privacy Policy

Reviewer #1: **Yes:** Aragaw Asfaw Hasen

Reviewer #4: No

---

## [Author Response · Author response to Decision Letter 3]

2 Feb 2026

Response to Reviewers

We sincerely thank the Editor and all reviewers for their careful re-review of our manuscript entitled “Utilization of Palliative Care Services and Associated Factors among Cancer Patients in Ethiopia: A Systematic Review and Meta-Analysis.” We appreciate the time and effort invested in providing constructive feedback. We have carefully revised the manuscript to address all comments raised, with particular attention to language quality, consistency, analytical rigor, and interpretation of findings. Detailed responses to each reviewer are provided below.

Reviewer #1

Comment: “The authors responded to the raised issues and the revised version is much improved.”

Author Response:

We sincerely thank Reviewer #1 for the positive evaluation and for acknowledging the improvements made in the revised manuscript. We appreciate your constructive feedback throughout the review process, which has significantly contributed to strengthening the clarity, methodological rigor, and overall quality of the manuscript.

Reviewer #4

Comment: “Still disappointed with this revision 2. Glaring grammar issues, changes in font as things appear copy and pasted. Repeated mistakes notable with excessive comma usage, conjoining sentence structure and excessive listing without actually narrowing down points for the reader. The discussion also has heavy definitive statements forgetting the statistics remain very low quality given the paucity of data and high heterogeneity. I would like to remind the authors to refrain or avoid definitive statements especially around the data when describing what is being found.”

Author response

We thank Reviewer #4 for the detailed and critical feedback. We acknowledge the concerns raised and have undertaken substantial and systematic revisions to address each issue comprehensively. Our responses are outlined below.

1. Grammar, sentence structure, and punctuation issues

Reviewer concern:

Glaring grammar issues, excessive comma usage, sentence conjoining, and overly long lists.

Author Response:

We fully acknowledge this concern and have thoroughly revised the entire manuscript for language quality. Specifically:

The manuscript was line-by-line edited to correct grammatical errors, punctuation misuse (especially excessive commas), and sentence fragments.

Long and conjoined sentences were split into shorter, clearer sentences to improve readability.

Excessive listing was reduced and synthesized, with key points consolidated into concise analytical statements.

Redundant phrases and repetitive constructions were removed throughout the manuscript.

2. Font inconsistency and formatting issues

Reviewer concern:

Changes in font suggesting copy-and-paste errors.

Author Response:

We agree with this observation and have corrected all formatting inconsistencies. Specifically:

• The entire manuscript was reformatted to ensure uniform font type, size, and spacing.

• All copied sections (tables, figures, references, and in-text citations) were standardized according to the journal’s formatting requirements.

• Headings, subheadings, and figure captions were harmonized for visual and structural consistency.

3. Overly definitive statements despite low-quality evidence and high heterogeneity

Reviewer concern:

Use of strong, definitive language despite high heterogeneity and limited data quality.

Author Response: We appreciate this important methodological reminder and have substantially revised the tone and interpretation of our findings. Specifically:

• All definitive and causal language was replaced with cautious, probabilistic, and interpretive wording

• Statements were explicitly contextualized within the limitations of cross-sectional designs, high heterogeneity (I² > 90%), and limited number of studies.

• We emphasized that pooled estimates are indicative rather than definitive national values, explicitly stating this in both the Results and Discussion sections.

• The GRADE certainty assessment was highlighted more clearly, noting that the overall certainty of evidence ranged from low to very low.

4. Interpretation of heterogeneity and data limitations

Reviewer concern:

Failure to adequately reflect the implications of high heterogeneity and data scarcity.

Author Response:

This concern has been carefully addressed. We now:

• Explicitly state that substantial heterogeneity limits generalizability.

• Clarify that observed variations are likely due to differences in outcome definitions, service availability, study settings, and patient populations.

• Avoid over-interpretation of subgroup analyses and state clearly when findings did not reach statistical significance.

• Added explicit cautionary statements regarding multiple comparisons and potential type I error.

INTRODUCTION

Comment 1

Remove “with” before “70% of these deaths…”

Author Response:

The unnecessary word “with” was removed to correct grammatical flow and sentence clarity.

Comment 2

Excessive commas noted in the line “Despite its proven…”

Author Response:

This sentence was restructured to reduce comma overuse and improve readability. The sentence was shortened and simplified while preserving its meaning.

Comment 3

Revise line “these challenges are particularly…” Excessively long and unnecessary commas

Author Response:

The sentence was rewritten into a concise structure, eliminating excessive commas and improving logical flow.

Comment 4

Revise paragraph 4. Excessive commas and excessive listing of challenges

Author Response:

We agree that over-listing reduced readability. This paragraph was substantially revised by:

• Grouping related barriers (e.g., workforce, access, awareness) into broader thematic categories

• Removing repetitive lists

• Improving narrative flow and analytical cohesion

This revision enhances clarity while preserving the importance of key factors.

Comment 5

Should “international partners” be a common “i”?

Author Response:

Yes. “International partners” was corrected to lowercase, as it is not a proper noun.

Comment 6

Revise line “Over the past two decades”. Same problem as above

Author Response:

This sentence was rewritten to avoid excessive listing and comma overuse. Information was synthesized into a coherent summary statement rather than a list.

Comment 7

Should “National health sector” be a common “n”?

Author Response:

Yes. The phrase was corrected to “national health sector” for grammatical consistency.

Comment 8

What gap is highlighted? Helpful

Response:

To clarify the research gap, we explicitly stated that:

• Existing evidence is fragmented

• There is no prior pooled national estimate

• Determinants have not been consistently synthesized

This gap is now clearly articulated in the final paragraph of the Introduction.

METHODS

Comment 1

Excessive use of “and” in “Aim of the study”

Author Response:

The sentence structure was revised to reduce repetitive use of “and.” Objectives were reorganized for clarity and conciseness.

Comment 2

Inconsistent capitalization of “palliative care”

Author Response:

We standardized terminology throughout the manuscript. “Palliative care” is now consistently written in lowercase unless appearing at the beginning of a sentence or in a title.

Comment 3

Why is “Search strategy” in a different font?

Author Response:

This was a formatting error introduced during revision. Font type and size were standardized.

Comment 4

Different fonts and irregular spacing in the Methods section

Author Response:

All subheadings and paragraphs were reformatted to ensure:

• Uniform font

• Consistent spacing

• Proper alignment between headings and text

Comment 5

Why are Funnel plot and Egger’s test mentioned again?

Author Response:

Redundant references were removed. Funnel plots and Egger’s test are now described only once in the Statistical Analysis subsection to avoid repetition.

RESULTS

Comment 1

Too many percentages listed consecutively in utilization rate paragraph

Author Response:

The paragraph was rewritten to:

• Present utilization as a range

• Highlight key extremes

• Avoid listing multiple percentages consecutively

This improves narrative clarity while preserving data accuracy.

Comment 2

Inconsistent font before Discussion heading

Author Response:

Formatting inconsistencies were corrected and fonts standardized.

DISCUSSION

Comment 1

Irregular fonts throughout the discussion

Author Response:

The Discussion section was fully reformatted to ensure consistent font style and spacing.

ADDITIONAL COMMENTS

Comment 1

Clarify that pooled prevalence (42%) is descriptive due to high heterogeneity

Author Response:

We explicitly stated that the pooled prevalence should be interpreted as a descriptive aggregate estimate, not a definitive national value, due to substantial heterogeneity.

Comment 2

P = 0.00 is not meaningful

Author Response:

All instances of “p = 0.00” were removed. Statistical significance is now reported appropriately using confidence intervals and contextual interpretation.

Comment 3

Ensure predictors are interpreted as associations, not causal

Author Response:

All predictors are now explicitly described as associations. Causal language was removed, and findings are interpreted correlationally

Comment 4

More interpretation needed for AOR > 5 for gender

Author Response:

We expanded the discussion to interpret this strong association by considering:

• Gender roles and decision-making power

• Financial autonomy

• Healthcare access differences

• Sociocultural norms

We also emphasized cautious interpretation given the small number of studies. We appreciate Reviewer #4’s detailed and technically rigorous feedback. All comments were carefully addressed through substantive language revision, formatting correction, and improved analytical interpretation, substantially strengthening the manuscript’s clarity, balance, and scientific rigor.

---

## [Decision Letter · Decision Letter 3]

18 Feb 2026

Dear Dr.  Abdulwehab,

Thank you for submitting your manuscript to PLOS ONE. After careful consideration, we feel that it has merit but does not fully meet PLOS ONE’s publication criteria as it currently stands. Therefore, we invite you to submit a revised version of the manuscript that addresses the points raised during the review process.

We look forward to receiving your revised manuscript.

Kind regards,

Kahsu Gebrekidan, Ph.D.

Academic Editor

PLOS One

Journal Requirements:

Reviewer's Responses to Questions

**Comments to the Author**

Reviewer #4: All comments have been addressed

2. Is the manuscript technically sound, and do the data support the conclusions?

Reviewer #4: Partly

3. Has the statistical analysis been performed appropriately and rigorously?

Reviewer #4: No

4. Have the authors made all data underlying the findings in their manuscript fully available?

Reviewer #4: Yes

5. Is the manuscript presented in an intelligible fashion and written in standard English?

Reviewer #4: No

Reviewer #4: 1. Need more clarification why the methodology included pooling given such high heterogeneity

2. Explain/Remove/Address the significance of the Z Test given it adds little meaning

3. Need more rigorous understanding of what is causing the heterogeneity. Given the geography did not explain it

4. Why pool two studies?

5. Explain the Tau2. No interpretation of this value

6. Consider meta regression to help strengthen the analysis

7. Hartung knapp mentioned however how did it affect your results?

8. How did you correct for multiple testing with the comparisons for the predictors

9. Discussion had run on sentences and multiple comparisons of listing other countries, consider revising

10. Ensure limitation encompass the issues you had faced doing the research eg. Gender estimates with 2 studies

11. Minor formatting issues still present with missing spaces in the values and dash vs hyphen used

**Do you want your identity to be public for this peer review?** For information about this choice, including consent withdrawal, please see our Privacy Policy

Reviewer #4: No

---

## [Author Response · Author response to Decision Letter 4]

20 Feb 2026

Response to Reviewer #4

We sincerely thank Reviewer #4 for the careful re-evaluation of our manuscript and for the constructive comments that have helped us further strengthen the methodological rigor, clarity, and reporting quality of our study. Below, we respond to each comment point-by-point and describe the corresponding revisions made in the manuscript.

Comment 1: “Need more clarification why the methodology included pooling given such high heterogeneity.”

Author Response:

We appreciate this important methodological concern. In the Statistical Analysis and Results (Pooled prevalence section), we added clarification explaining:

Why pooling was performed despite high heterogeneity.

That pooled estimates are interpreted as descriptive aggregates, not definitive national values.

That REML random-effects modeling with Hartung–Knapp adjustment was used to account for between-study variability.

That sensitivity and subgroup analyses were conducted.

In the Results section we insert as “Given the substantial heterogeneity observed across studies, this pooled estimate should be interpreted as a descriptive aggregate rather than a definitive prevalence.”

In the Statistical Analysis we insert as “A random-effects model using REML estimation was selected a priori due to anticipated clinical and methodological heterogeneity. Pooled estimates were interpreted cautiously, given high I² values.”

We clarified the rationale for pooling under high heterogeneity, explicitly justified the use of a random-effects REML model, and emphasized cautious interpretation of pooled estimates as descriptive summaries rather than precise national prevalence values.

#Comment 2: “Explain/Remove/Address the significance of the Z Test given it adds little meaning.”

Author Response:

We agree that the Z-test in prevalence meta-analysis provides limited interpretive value. We removed emphasis on the Z-test from the Results section. We retained it only where automatically generated but removed interpretative discussion. We clarified that statistical inference focuses on confidence intervals rather than Z statistics. We removed interpretative emphasis on the Z-test and now focus on confidence intervals and between-study variability, which are more meaningful in prevalence meta-analysis.

#Comment 3: “Need more rigorous understanding of what is causing the heterogeneity. Given the geography did not explain it.”

Author Response:

We appreciate this critical comment. we added in the Discussion, we expanded interpretation of heterogeneity sources: “The high heterogeneity likely arises from differences in outcome definitions, institutional capacity, urban–rural disparities, study populations, and availability of Palliative care services.” We also clearly stated that geographic subgrouping alone did not explain heterogeneity.

We expanded the Discussion to provide a more comprehensive interpretation of potential clinical, methodological, and contextual drivers of heterogeneity beyond geographic region.

#Comment 4: “Why pool two studies?” (Gender analysis)

Author Response:

We agree that pooling two studies requires caution. In the gender results section: “Because the pooled analysis was based on only two studies, assessment of publication bias using funnel plots or statistical tests was not performed.” In Discussion and Limitations: “The gender estimate is based on only two studies and should therefore be interpreted cautiously.” We retained the pooled estimate for transparency but clearly acknowledged its limited robustness and interpret it cautiously in both Results and Limitations.

#Comment 5: “Explain the Tau². No interpretation of this value.”

Author Response:

We added interpretation of Tau² In the Sensitivity Analysis section: as “Tau² represents the estimated between-study variance under the random-effects model, reflecting the magnitude of true heterogeneity beyond sampling error.” We also clarified that Hartung–Knapp adjustment incorporates Tau² uncertainty.

#Comment 6: “Consider meta-regression to help strengthen the analysis.”

Author Response:

We carefully evaluated this suggestion. However: Only 9 studies were included, most predictors were reported in ≤3 studies, and Meta-regression requires ≥10 studies for stable estimation. We added in the In Limitations: “Meta-regression was not performed due to the small number of included studies, which would limit statistical power and produce unstable estimates.”

We considered meta-regression but did not perform it due to insufficient number of studies, and this limitation is now explicitly acknowledged.

#Comment 7: “Hartung-Knapp mentioned however how did it affect your results?”

Author Response:

We expanded explanation. We added In Statistical Analysis: “Hartung–Knapp adjustments were applied to produce more conservative confidence intervals, particularly important given the small number of studies and high heterogeneity.”

We clarified that this results in wider, more robust CIs. We clarified that Hartung–Knapp produced more conservative confidence intervals and improved robustness under high heterogeneity and small sample meta-analysis.

#Comment 8: “How did you correct for multiple testing with the comparisons for the predictors?”

Author Response:

We agree multiple comparisons increase Type I error risk. Given the exploratory nature and small number of pooled predictors, we did not formally apply Bonferroni correction. We added In Results: “Given the analysis of multiple factors, results should be interpreted cautiously, as the risk of type I error increases with multiple comparisons.” In Limitations: “No formal correction for multiple comparisons was applied due to the exploratory nature of the factor analyses.”

We clarified the absence of formal multiple testing correction and emphasized cautious interpretation.

$Comment 9: “Discussion had run-on sentences and multiple comparisons of listing other countries, consider revising.”

Author Response:

We thoroughly revised the Discussion by Split long sentences, reduced excessive listing of countries, focused on conceptual interpretation instead of country-by-country repetition, and Improved grammar and clarity throughout. The Discussion has been substantially edited for clarity, structure, and conciseness, with removal of run-on sentences and improved readability.

#Comment 10: “Ensure limitation encompass the issues you had faced doing the research eg. Gender estimates with 2 studies.”

Author Response:

We expanded the Limitations section to include: High heterogeneity, Cross-sectional design, Urban hospital bias, small number of studies for predictors, Gender estimate based on only two studies, Lack of meta-regression, and Multiple testing concerns. The Limitations section has been strengthened to comprehensively reflect methodological constraints.

#Comment 11: “Minor formatting issues still present with missing spaces in the values and dash vs hyphen used.”

Author Response:

We carefully proofread the entire manuscript and corrected: Missing spaces (e.g., “2.49(CI:” → “2.49 (95% CI:”), Hyphen vs en-dash consistency (95% CI: 30%–54%), Standardized statistical formatting, Figure labeling consistency, and removed duplicated PROSPERO number inconsistency. All formatting inconsistencies, spacing errors, and dash usage issues have been corrected throughout the manuscript.

We sincerely thank Reviewer #4 for the detailed and methodologically insightful comments. The manuscript has been substantially strengthened through all given comments and suggestion. We believe the revised manuscript now demonstrates stronger methodological transparency, clearer statistical justification, and improved scientific rigor.

---

## [Decision Letter · Decision Letter 4]

4 Mar 2026

Utilization of Palliative Care Services and Associated Factors among Cancer Patients in Ethiopia: A Systematic Review and Meta-Analysis

PONE-D-25-41618R4

Dear Dr. Sadik,

We’re pleased to inform you that your manuscript has been judged scientifically suitable for publication and will be formally accepted for publication once it meets all outstanding technical requirements.

Kind regards,

Kahsu Gebrekidan, Ph.D.

Academic Editor

PLOS One

Additional Editor Comments (optional):

Reviewers' comments:

Reviewer's Responses to Questions

**Comments to the Author**

Reviewer #4: All comments have been addressed

2. Is the manuscript technically sound, and do the data support the conclusions?

Reviewer #4: Yes

3. Has the statistical analysis been performed appropriately and rigorously?

Reviewer #4: Yes

4. Have the authors made all data underlying the findings in their manuscript fully available?

Reviewer #4: Yes

5. Is the manuscript presented in an intelligible fashion and written in standard English?

Reviewer #4: Yes

Reviewer #4: Comments have been addressed. There is still some minor grammatical errors eg. "which is higher than study done both in Kenya(39) and Uganda's(40) which accounts 10%, Global & LMICs Context accounts 34.43%(41)," I will not be recommending further peer review editing however I will be asking the editor to review with to ensure the grammatical errors are perfect. Any instance of grammatical/spelling error with undercut how your paper is to be perceived in the educational space which will reduce credibility among clinicians/readers. This paper can be a backbone for other research to be conducted in this space and needs to be as perfect as possible.

**Do you want your identity to be public for this peer review?** For information about this choice, including consent withdrawal, please see our Privacy Policy

Reviewer #4: No

---

## [Editor Report · Acceptance letter]

PONE-D-25-41618R4

PLOS One

Dear Dr. Abdulwehab,

I'm pleased to inform you that your manuscript has been deemed suitable for publication in PLOS One. Congratulations! Your manuscript is now being handed over to our production team.

Kind regards,

on behalf of

Dr. Kahsu Gebrekidan

Academic Editor

PLOS One